# EmbedDistill: A Geometric Knowledge Distillation for Information Retrieval

## Abstract

Large neural models (such as Transformers) achieve state-of-the-art performance for information retrieval (IR). In this paper, we aim to improve distillation methods that pave the way for the resource-efficient deployment of such models in practice. Inspired by our theoretical analysis of the teacher-student generalization gap for IR models, we propose a novel distillation approach that leverages the relative geometry among queries and documents learned by the large teacher model. Unlike existing teacher score-based distillation methods, our proposed approach employs embedding matching tasks to provide a stronger signal to align the representations of the teacher and student models. In addition, it utilizes query generation to explore the data manifold to reduce the discrepancies between the student and the teacher where training data is sparse. Furthermore, our analysis also motivates novel asymmetric architectures for student models which realizes better embedding alignment without increasing online inference cost. On standard benchmarks like MSMARCO, we show that our approach successfully distills from both dual-encoder (DE) and cross-encoder (CE) teacher models to 1/10th size asymmetric students that can retain 95-97% of the teacher performance.

## 1 Introduction

Neural models for information retrieval (IR) are increasingly used to model the true ranking function in various applications, including web search [38], recommendation [65], and question-answering (QA) [6]. Notably, the recent success of Transformers [59]-based pre-trained language models [11, 30, 49] on a wide range of natural language understanding tasks has also prompted their utilization in IR to capture query-document relevance [see, e.g., 10, 34, 43, 26, 20].

A typical IR system comprises two stages: (1) A *retriever* first selects a small subset of potentially relevant candidate documents (out of a large collection) for a given query; and (2) A *re-ranker* then identifies a precise ranking among the candidates provided by the retriever. *Dual-encoder* (DE) models are the de-facto architecture for retrievers [26, 20]. Such models independently embed queries and documents into a common space, and capture their relevance by simple operations on these embeddings such as the inner product. This enables offline creation of a document index and supports fast retrieval during inference via efficient maximum inner product search implementations [12, 19], with *online* query embedding generation primarily dictating the inference latency. *Cross-encoder* (CE) models, on the other hand, are preferred as re-rankers, owing to their excellent performance [43, 9, 62]. A CE model jointly encodes a query-document pair while enabling early interaction among query and document features. Employing a CE model for retrieval is often infeasible, as it would require processing a given query with *every* document in the collection at inference time. In fact, even in the re-ranking stage, the inference cost of CE models is high enough [22] to warrant exploration of efficient alternatives [14, 22, 37]. Across both architectures, scaling to larger models brings improved performance at increased computational cost [41, 39].

Submitted to 37th Conference on Neural Information Processing Systems (NeurIPS 2023). Do not distribute.

*Knowledge distillation* [5, 13] provides a general strategy to address the prohibitive inference cost associated with high-quality large neural models. In the IR literature, most existing distillation methods only rely on the teacher's query-document relevance scores [see, e.g., 31, 14, 8, 51, 56] or their proxies [16]. However, given that neural IR models are inherently embedding-based, it is natural to ask: *Is it useful to go beyond matching of the teacher and student models'* scores, *and directly aim to align their* embedding spaces?

With this in mind, we propose a novel distillation method for IR models that utilizes an *embedding matching* task to train student models. The proposed method is inspired by our rigorous treatment of the generalization gap between the teacher and student models in IR settings. Our theoretical analysis of the *teacher-student generalization gap* further suggests novel design choices involving *asymmetric configurations* for student DE models, intending to further reduce the gap by better aligning teacher and student embedding spaces. Notably, our proposed distillation method supports *cross-architecture distillation* and improves upon existing (score-based) distillation methods for both retriever and re-ranker models. When distilling a large teacher DE model into a smaller student DE model, for a given query (document), one can minimize the distance between the query (document) embeddings of the teacher and student (after compatible projection layers to account for dimension mismatch, if any). In contrast, a teacher CE model doesn't directly provide document and query embeddings, and so to effectively employ embedding matching-based distillation requires modifying the scoring layer with *dual-pooling* [61] and adding various regularizers. Both of these changes improve geometry of teacher embeddings and facilitate effective knowledge transfer to the student DE model via embedding matching-based distillation.

Our key contributions toward improving IR models via distillation are:

- We provide the first rigorous analysis of the teacher-student generalization gap for IR settings which captures the role of alignment of embedding spaces of the teacher and student towards reducing the gap (Sec. 3).

- Inspired by our analysis, we propose a novel distillation approach for neural IR models, namely EmbedDistill, that goes beyond score matching and aligns the embedding spaces of the teacher and student models (Sec. 4). We also show that EmbedDistill can leverage synthetic data to improve a student by further aligning the embedding spaces of the teacher and student (Sec. 4.3).

- Our analysis motivates novel distillation setups. Specifically, we consider a student DE model with an *asymmetric* configuration, consisting of a small query encoder and a *frozen* document encoder inherited from the teacher. This significantly reduces inference latency of query embedding generation, while leveraging the teachers' high-quality document index (Sec. 4.1).

- We provide a *comprehensive* empirical evaluation of EmbedDistill (Sec. 5) on two standard IR benchmarks – Natural Questions [23] and MSMARCO [40]. We also evaluate EmbedDistill on BEIR benchmark [57] which is used to measure the *zero-shot* performance of an IR model.

Note that prior works have utilized embedding alignment during distillation for *non-IR* setting [see, e.g., 52, 55, 18, 1, 64, 7]. However, to the best of our knowledge, our work is the first to study embedding matching-based distillation method for IR settings which requires addressing multiple IR-specific challenges such as cross-architecture distillation, partial representation alignment, and enabling novel asymmetric student configurations. Furthermore, unlike these prior works, our proposed method is theoretically justified to reduce the teacher-student performance gap.

## 2 Background

Let $\mathcal{Q}$ and $\mathcal{D}$ denote the query and document spaces, respectively. An IR model is equivalent to a scorer $s : \mathcal{Q} \times \mathcal{D} \to \mathbb{R}$, i.e., it assigns a (relevance) score $s(q, d)$ for a query-document pair $(q, d) \in \mathcal{Q} \times \mathcal{D}$. Ideally, we want to learn a scorer such that $s(q, d) > s(q, d')$ *iff* the document $d$ is more relevant to the query $q$ than document $d'$. We assume access to $n$ labeled training examples $\mathcal{S}_n = \{(q_i, \mathbf{d}_i, \mathbf{y}_i)\}_{i \in [n]}$. Here, $\mathbf{d}_i = (d_{i,1}, \ldots, d_{i,L}) \in \mathcal{D}^L$, $\forall i \in [n]$, denotes a list of $L$ documents and $\mathbf{y}_i = (y_{i,1}, \ldots, y_{i,L}) \in \{0, 1\}^L$ denotes the corresponding labels such that $y_{i,j} = 1$ iff the document $d_{i,j}$ is relevant to the query $q_i$. Given $\mathcal{S}_n$, we learn an IR model by minimizing

$$R(s; \mathcal{S}_n) := \frac{1}{n} \sum_{i \in [n]} \ell\big(s_{q_i, \mathbf{d}_i}, \mathbf{y}_i\big), \tag{1}$$

where $s_{q_i, \mathbf{d}_i} := (s(q_i, d_{1,i}), \ldots, s(q_i, d_{1,L}))$ and $\ell\big(s_{q_i, \mathbf{d}_i}, \mathbf{y}_i\big)$ denotes the loss $s$ incurs on $(q_i, \mathbf{d}_i, \mathbf{y}_i)$. Due to space constraint, we defer concrete choices for the loss function $\ell$ to Appendix A.

While this learning framework is general enough to work with any IR models, next, we formally introduce two families of Transformer-based IR models that are prevalent in the recent literature.

## 2.1 Transformer-based IR models: Cross-encoders and Dual-encoders

Let query $q = (q^1, \ldots, q^{m_1})$ and document $d = (d^1, \ldots, d^{m_2})$ consist of $m_1$ and $m_2$ tokens, respectively. We now discuss how Transformers-based CE and DE models process the $(q, d)$ pair.

**Cross-encoder model.** Let $p = [q; d]$ be the sequence obtained by concatenating $q$ and $d$. Further, let $\tilde{p}$ be the sequence obtained by adding special tokens such $[\texttt{CLS}]$ and $[\texttt{SEP}]$ to $p$. Given an encoder-only Transformer model $\mathrm{Enc}$, the relevance score for the $(q, d)$ pair is

$$s(q, d) = \langle w, \mathrm{pool}\big(\mathrm{Enc}(\tilde{p})\big)\rangle = \langle w, \texttt{emb}_{q,d}\rangle, \tag{2}$$

where $w$ is a $d$-dimensional classification vector, and $\mathrm{pool}(\cdot)$ denotes a pooling operation that transforms the contextualized token embeddings $\mathrm{Enc}(\tilde{p})$ to a joint embedding vector $\texttt{emb}_{q,d}$. $[\texttt{CLS}]$-pooling is a common operation that simply outputs the embedding of the $[\texttt{CLS}]$ token as $\texttt{emb}_{q,d}$.

**Dual-encoder model.** Let $\tilde{q}$ and $\tilde{d}$ be the sequences obtained by adding appropriate special tokens to $q$ and $d$, respectively. A DE model comprises two (encoder-only) Transformers $\mathrm{Enc}_Q$ and $\mathrm{Enc}_D$, which we call query and document encoders, respectively.[1] Let $\texttt{emb}_q = \mathrm{pool}\big(\mathrm{Enc}_Q(\tilde{q})\big)$ and $\texttt{emb}_d = \mathrm{pool}\big(\mathrm{Enc}_D(\tilde{d})\big)$ denote the query and document embeddings, respectively. Now, one can define $s(q, d) = \langle \texttt{emb}_q, \texttt{emb}_d \rangle$ to be the relevance score assigned to the $(q, d)$ pair by the DE model.

## 2.2 Score-based distillation for IR models

Most distillation schemes for IR [e.g., 31, 14, 8] rely on teacher relevance scores. Given a training set $\mathcal{S}_n$ and a teacher with scorer $s^\mathrm{t}$, one learns a student with scorer $s^\mathrm{s}$ by minimizing

$$R(s^\mathrm{s}, s^\mathrm{t}; \mathcal{S}_n) = \frac{1}{n} \sum\nolimits_{i \in [n]} \ell_\mathrm{d}\big(s^\mathrm{s}_{q, \mathbf{d}_i}, s^\mathrm{t}_{q, \mathbf{d}_i}\big), \tag{3}$$

where $\ell_\mathrm{d}$ captures the discrepancy between $s^\mathrm{s}$ and $s^\mathrm{t}$. See Appendix A for common choices for $\ell_\mathrm{d}$.

# 3 Teacher-student generalization gap: Inspiration for embedding alignment

Our main objective is to devise novel distillation methods to realize high-performing student DE models. As a first step in this direction, we rigorously study the teacher-student generalization gap as realized by standard (score-based) distillation in IR settings. Informed by our analysis, we subsequently identify novel ways to improve the student model's performance. In particular, our analysis suggests two natural directions to reduce the teacher-student generalization gap: 1) enforcing tighter alignment between embedding spaces of teacher and student models; and 2) exploring novel asymmetric configuration for student DE model.

Let $R(s) = \mathbb{E}\big[\ell\big(s_{q, \mathbf{d}}, \mathbf{y}\big)\big]$ be the population version of the empirical risk in Eq. 1, which measures the test time performance of the IR model defined by the scorer $s$. Thus, $R(s^\mathrm{s}) - R(s^\mathrm{t})$ denotes the *teacher-student generalization gap*. In the following result, we bound this quantity (see Appendix C.1 for a formal statement and proof). We focus on distilling a teacher DE model to a student DE model and $L = 1$ (cf. Sec. 2) as it leads to easier exposition without changing the main takeaways. Our analysis can be extended to $L > 1$ or CE to DE distillation with more complex notation.

**Theorem 3.1** (Teacher-student generalization gap (informal)). *Let $\mathcal{F}$ and $\mathcal{G}$ denote the function classes for the query and document encoders for the student model, respectively. Suppose that the score-based distillation loss $\ell_\mathrm{d}$ in Eq. 3 is based on binary cross entropy loss (Eq. 12 in Appendix A). Let one-hot (label-dependent) loss $\ell$ in Eq. 1 be the binary cross entropy loss (Eq. 10 in Appendix A). Further, assume that all encoders have the same output dimension and embeddings have their $\ell_2$-norm bounded by $K$. Then, we have*

$$R(s^\mathrm{s}) - R(s^\mathrm{t}) \leq \mathcal{E}_n(\mathcal{F}, \mathcal{G}) + 2K R_{\mathrm{Emb}, Q}(\mathrm{t}, \mathrm{s}; \mathcal{S}_n) + 2K R_{\mathrm{Emb}, D}(\mathrm{t}, \mathrm{s}; \mathcal{S}_n)$$
$$+ \Delta(s^\mathrm{t}; \mathcal{S}_n) + K^2\big(\mathbb{E}\big[\big|\sigma(s^\mathrm{t}_{q,d}) - y\big|\big] + \frac{1}{n} \sum_{i \in [n]} \big|\sigma(s^\mathrm{t}_{q_i, d_i}) - y_i\big|\big), \tag{4}$$

---

[1] It is common to employ dual-encoder models where query and document encoders are shared.

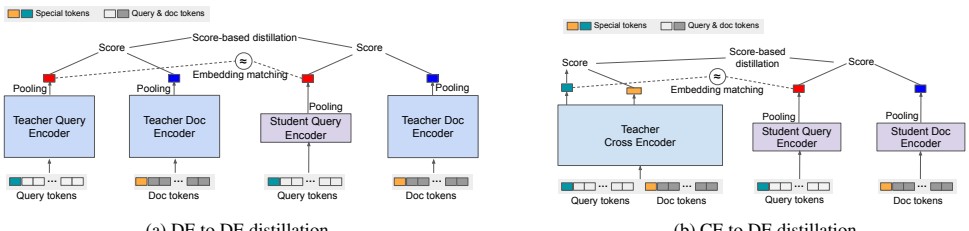

(a) DE to DE distillation          (b) CE to DE distillation

Figure 1: Proposed distillation method with query embedding matching. **Left:** The setting where student employs an asymmetric DE configuration with a small query encoder and a large (non-trainable) document encoder inherited from the teacher DE model. The smaller query encoder ensures small latency for encoding query during inference, and large document encoder leads to a good quality document index. **Right:** Similarly the setting of CE to DE distillation using EmbedDistill, with teacher CE model employing dual pooling.

130   *where* $\mathcal{E}_n(\mathcal{F}, \mathcal{G}) := \sup_{s^{\mathrm{s}} \in \mathcal{F} \times \mathcal{G}} \left| R(s^{\mathrm{s}}, s^{\mathrm{t}}; \mathcal{S}_n) - \mathbb{E}\ell_{\mathrm{d}}\left(s^{\mathrm{s}}_{q,d}, s^{\mathrm{t}}_{q,d}\right) \right|$; $\sigma$ *denotes the sigmoid function;*
131   *and* $\Delta(s^{\mathrm{t}}; \mathcal{S}_n)$ *denotes the deviation between the empirical risk (on* $\mathcal{S}_n$*) and population risk of the*
132   *teacher* $s^{\mathrm{t}}$*. Here,* $R_{\mathrm{Emb},Q}(\mathrm{t}, \mathrm{s}; \mathcal{S}_n)$ *and* $R_{\mathrm{Emb},D}(\mathrm{t}, \mathrm{s}; \mathcal{S}_n)$ *measure misalignment between teacher and*
133   *student embeddings by focusing on queries and documents, respectively (cf. Eq. 7 & 8 in Sec. 4.1).*

134   The last three quantities in the bound in Thm. 3.1, namely $\Delta(s^{\mathrm{t}}; \mathcal{S}_n)$, $\mathbb{E}[|\sigma(s^{\mathrm{t}}_{q,d}) - y|]$, and
135   $\frac{1}{n}\sum_{i \in [n]} |\sigma(s^{\mathrm{t}}_{q_i,d_i}) - y_i|$, are *independent* of the underlying student model. These terms solely
136   depend on the quality of the underlying teacher model $s^{\mathrm{t}}$. That said, the teacher-student gap can be
137   made small by reducing the following three terms: 1) uniform deviation of the student's empirical
138   distillation risk from its population version $\mathcal{E}_n(\mathcal{F}, \mathcal{G})$; 2) misalignment between teacher student query
139   embeddings $R_{\mathrm{Emb},Q}(\mathrm{t}, \mathrm{s}; \mathcal{S}_n)$; and 3) misalignment between teacher student document embeddings
140   $R_{\mathrm{Emb},D}(\mathrm{t}, \mathrm{s}; \mathcal{S}_n)$.

141   The last two terms motivate us to propose an *embedding matching*-based distillation that explicitly
142   aims to minimize these terms during student training. Even more interestingly, these terms also
143   inspire an *asymmetric DE configuration* for the student which strikes a balance between the goals of
144   reducing the misalignment between the embeddings of teacher and student (by inheriting teacher's
145   document encoder) and ensuring serving efficiency (small inference latency) by employing a small
146   query encoder. Before discussing these proposals in detail in Sec. 4 and Fig. 1, we explore the first
147   term $\mathcal{E}_n(\mathcal{F}, \mathcal{G})$ and highlight how our proposals also have implications for reducing this term. Towards
148   this, the following result bounds $\mathcal{E}_n(\mathcal{F}, \mathcal{G})$. Due to space constraints, we present an informal statement
149   of the result (see Appendix C.2 for a more precise statement and proof).

150   **Proposition 3.2.** *Let* $\ell_{\mathrm{d}}$ *be a distillation loss which is* $L_{\ell_{\mathrm{d}}}$*-Lipschitz in its first argument. Let* $\mathcal{F}$ *and* $\mathcal{G}$
151   *denote the function classes for the query and document encoders, respectively. Further assume that,*
152   *for each query and document encoder in our function class, the query and document embeddings*
153   *have their* $\ell_2$*-norm bounded by* $K$*. Then,*

$$\mathcal{E}_n(\mathcal{F}, \mathcal{G}) \leq \mathbb{E}_{\mathcal{S}_n} \frac{48 K L_{\ell_{\mathrm{d}}}}{\sqrt{n}} \int_0^\infty \sqrt{\log\left(N(u, \mathcal{F}) N(u, \mathcal{G})\right)}\, du. \qquad (5)$$

154   *Furthermore, with a fixed document encoder, i.e.,* $\mathcal{G} = \{g^*\}$,

$$\mathcal{E}_n(\mathcal{F}, \{g*\}) \leq \mathbb{E}_{\mathcal{S}_n} \frac{48 K L_{\ell_{\mathrm{d}}}}{\sqrt{n}} \int_0^\infty \sqrt{\log N(u, \mathcal{F})}\, du. \qquad (6)$$

155   *Here,* $N(u, \cdot)$ *is the* $u$*-covering number of a function class.*

156   Note that Eq. 5 and Eq. 6 correspond to uniform deviation when we train *without* and *with* a frozen
157   document encoder, respectively. It is clear that the bound in Eq. 6 is less than or equal to that in
158   Eq. 5 (because $N(u, \mathcal{G}) \geq 1$ for any $u$), which alludes to desirable impact of employing a frozen
159   document encoder as one of our proposal seeks to do via *inheriting teacher's document encoder* (for
160   instance in an asymmetric DE configuration). Furthermore, our proposal of employing an embedding-
161   matching task will regularize the function class of query encoders; effectively reducing it to $\mathcal{F}'$ with
162   $|\mathcal{F}'| \leq |\mathcal{F}|$. The same holds true for document encoder function class when document encoder is
163   trainable (as in Eq. 5), leading to an effective function class $\mathcal{G}'$ with $|\mathcal{G}'| \leq |\mathcal{G}|$. Since we would have
164   $N(u, \mathcal{F}') \leq N(u, \mathcal{F})$ and $N(u, \mathcal{G}') \leq N(u, \mathcal{G})$, this suggests desirable implications of embedding
165   matching for reducing the uniform deviation bound.

## 4 Embedding-matching based distillation

Informed by our analysis of teacher-student generalization gap in Sec. 3, we propose EmbedDistill – a novel distillation method that explicitly focuses on aligning the embedding spaces of the teacher and student. Our proposal goes beyond existing distillation methods in the IR literature that only use the teacher scores. Next, we introduce EmbedDistill for two prevalent settings: (1) distilling a large DE model to a smaller DE model; [2] and (2) distilling a CE model to a DE model.

### 4.1 DE to DE distillation

Given a $(q, d)$ pair, let $\mathsf{emb}_q^t$ and $\mathsf{emb}_d^t$ be the query and document embeddings produced by the query encoder $\mathrm{Enc}_Q^t$ and document encoder $\mathrm{Enc}_D^t$ of the teacher DE model, respectively. Similarly, let $\mathsf{emb}_q^s$ and $\mathsf{emb}_d^s$ denote the query and document embeddings produced by a student DE model with $(\mathrm{Enc}_Q^s, \mathrm{Enc}_D^s)$ as its query and document encoders. Now, EmbedDistill optimizes the following embedding alignment losses in addition to the score-matching loss from Sec. 2.2 to align query and document embeddings of the teacher and student:

$$R_{\mathrm{Emb},Q}(\mathrm{t},\mathrm{s};\mathcal{S}_n) = \frac{1}{n}\sum\nolimits_{q\in\mathcal{S}_n} \|\mathsf{emb}_q^t - \mathrm{proj}(\mathsf{emb}_q^s)\|; \tag{7}$$

$$R_{\mathrm{Emb},D}(\mathrm{t},\mathrm{s};\mathcal{S}_n) = \frac{1}{n}\sum\nolimits_{d\in\mathcal{S}_n} \|\mathsf{emb}_d^t - \mathrm{proj}(\mathsf{emb}_d^s)\|. \tag{8}$$

**Asymmetric DE.** We also propose a novel student DE configuration where the student employs the teacher's document encoder (i.e., $\mathrm{Enc}_D^s = \mathrm{Enc}_D^t$) and only train its query encoder, which is much smaller compared to the teacher's query encoder. For such a setting, it is natural to only employ the embedding matching loss in Eq. 7 as the document embeddings are aligned by design (cf. Fig. 1a).

Note that this asymmetric student DE does not incur an increase in latency despite the use of a large teacher document encoder. This is because the large document encoder is only needed to create a good quality document index offline, and only the query encoder is evaluated at inference time. Also, the similarity search cost is not increased as the projection layer ensures the same small embedding dimension as in the symmetric DE student. Thus, for DE to DE distillation, we prescribe the asymmetric DE configuration universally. Our theoretical analysis (cf. Sec. 3) and experimental results (cf. Sec. 5) suggest that the ability to inherit the document tower from the teacher DE model can drastically improve the final performance, especially when combined with query embedding matching task (cf. Eq. 7).

### 4.2 CE to DE distillation

Given that CE models jointly encode query-document pairs, individual query and document embeddings are not readily available to implement embedding matching losses as per Eq. 7 and 8. This makes it challenging to employ EmbedDistill for CE to DE distillation.

As a naïve solution, for a $(q, d)$ pair, one can simply match a joint transformation of the student's query embedding $\mathsf{emb}_q^s$ and document embedding $\mathsf{emb}_d^s$ to the teacher's joint embedding $\mathsf{emb}_{q,d}^t$, produced by (single) teacher encoder $\mathrm{Enc}^t$. However, we observed that including such an embedding matching task often leads to severe over-fitting, and results in a poor student. Since $s^t(q, d) = \langle w, \mathsf{emb}_{q,d}^t\rangle$, during CE model training, the joint embeddings $\mathsf{emb}_{q,d}^t$ for relevant and irrelevant $(q, d)$ pairs are encouraged to be aligned with $w$ and $-w$, respectively. This produces degenerate embeddings that do not capture semantic query-to-document relationships. We notice that even the final query and document token embeddings lose such semantic structure (cf. Appendix G.2). Thus, a teacher CE model with $s^t(q, d) = \langle w, \mathsf{emb}_{q,d}^t\rangle$ does not add value for distillation beyond score-matching; in fact, it *hurts* to include naïve embedding matching. Next, we propose a modified CE model training strategy that facilitates EmbedDistill.

**CE models with dual pooling.** A *dual pooling* scheme is employed in the scoring layer to produce two embeddings $\mathsf{emb}_{q\leftarrow(q,d)}^t$ and $\mathsf{emb}_{d\leftarrow(q,d)}^t$ from a CE model that serve as the *proxy* query and document embeddings, respectively. Accordingly, we define the relevance score as $s^t(q, d) = \langle\mathsf{emb}_{q\leftarrow(q,d)}^t, \mathsf{emb}_{d\leftarrow(q,d)}^t\rangle$. We explore two variants of dual pooling: (1) special token-based pooling that pools from [CLS] and [SEP]; and (2) segment-based weighted mean pooling that separately

---

[2]CE to CE distillation is a special case of this with classification vector $w$ (cf. Eq. 2) as trivial second encoder.

Table 1: *Full* recall performance of various student DE models on NQ dev set, including symmetric DE student model (67.5M or 11.3M transformer for both encoders), and asymmetric DE student model (67.5M or 11.3M transformer as query encoder and document embeddings inherited from the teacher). All distilled students used the same teacher (110.1M parameter BERT-base models as both encoders), with the full Recall@5 = 72.3, Recall@20 = 86.1, and Recall@100 = 93.6.

| Method | 6-Layer (67.5M) | | | 4-Layer (11.3M) | | |
|---|---|---|---|---|---|---|
| | R@5 | R@20 | R@100 | R@5 | R@20 | R@100 |
| Train student directly | 36.2 | 59.7 | 80.0 | 24.8 | 44.7 | 67.5 |
| + Distill from teacher | 65.3 | 81.6 | 91.2 | 44.3 | 64.9 | 81.0 |
| + Inherit doc embeddings | 69.9 | 83.9 | 92.3 | 56.3 | 70.9 | 82.5 |
| + Query embedding matching | 72.7 | **86.5** | **93.9** | 61.2 | 75.2 | 85.1 |
| + Query generation | **73.4** | 86.3 | 93.8 | **64.3** | **77.8** | **87.9** |
| Train student using only embedding matching and inherit doc embeddings | 71.4 | 84.9 | 92.6 | 64.6 | 50.2 | 76.8 |
| + Query generation | 71.8 | 85.0 | 93.0 | 54.2 | 68.9 | 80.8 |

Table 2: Performance of EmbedDistill for DE to DE distillation on NQ test set. While prior works listed in the table rely on techniques such as negative mining and multi-stage training, we explore the orthogonal direction of embedding-matching that improves *single-stage* distillation, which can be combined with them.

| Method | #Layers | R@20 | R@100 |
|---|---|---|---|
| DPR [20] | 12 | 78.4 | 85.4 |
| DPR + PAQ [47] | 12 | 84.0 | 89.2 |
| DPR + PAQ [47] | 24 | 84.7 | 89.2 |
| ACNE [60] | 12 | 81.9 | 87.5 |
| RocketQA [48] | 12 | 82.7 | 88.5 |
| MSS-DPR [53] | 12 | 84.0 | 89.2 |
| MSS-DPR [53] | 24 | 84.8 | 89.8 |
| Our teacher [63] | 12 (220.2M) | 85.4 | 90.0 |
| EmbedDistill | 6 (67.5M) | 85.1 | 89.8 |
| EmbedDistill | 4 (11.3M) | 81.2 | 87.4 |

performs weighted averaging on the query and document segments of the final token embeddings. See Appendix B for details.

In addition to dual pooling, we also utilize a reconstruction loss during the CE training, which measures the likelihood of predicting each token of the original input from the final token embeddings. This loss encourages reconstruction of query and document tokens based on the final token embeddings and prevents the degeneration of the token embeddings during training. Given proxy embeddings from the teacher CE, we can perform EmbedDistill with the embedding matching loss defined in Eq. 7 and Eq. 8 (cf. Fig. 1b).

### 4.3 Task-specific online data generation

Data augmentation as a general technique has been previously considered in the IR literature [see, e.g., 45, 47, 17], especially in data-limited, out-of-domain, or zero-shot settings. As EmbedDistill aims to align the embeddings spaces of the teacher and student, the ability to generate similar queries or documents can naturally help enforce such an alignment globally on the task-specific manifold. Given a set of unlabeled task-specific query and document pairs $\mathcal{U}_m$, we can further add the embedding matching losses $R_{\text{Emb},\text{Q}}(\text{t}, \text{s}; \mathcal{U}_m)$ or $R_{\text{Emb},\text{D}}(\text{t}, \text{s}; \mathcal{U}_m)$ to our training objective. Interestingly, for DE to DE distillation setting, our approach can even benefit from a large collection of task-specific queries $\mathcal{Q}'$ or documents $\mathcal{D}'$. Here, we can independently employ embedding matching losses $R_{\text{Emb},\text{Q}}(\text{t}, \text{s}; \mathcal{Q}')$ or $R_{\text{Emb},\text{D}}(\text{t}, \text{s}; \mathcal{D}')$ that focus on queries and documents, respectively. Please refer to Appendix E describing how the task-specific data were generated.

## 5 Experiments

We now conduct a comprehensive evaluation of the proposed distillation approach. Specifically, we highlight the utility of the approach for both DE to DE and CE to DE distillation. We also showcase the benefits of combining our distillation approach with query generation methods.

### 5.1 Setup

**Benchmarks and evaluation metrics.** We consider two popular IR benchmarks — Natural Questions (NQ) [24] and MSMARCO [40], which focus on finding the most relevant passage/document given a question and a search query, respectively. NQ provides both standard test and dev sets, whereas MSMARCO provides only the dev set that are widely used for common benchmarks. In what follows, we use the terms query (document) and question (passages) interchangeably. For NQ, we use the standard full recall (*strict*) as well as the *relaxed* recall metric [20] to evaluate the retrieval performance. For MSMARCO, we focus on the standard metrics *Mean Reciprocal Rank* (MRR)@10, and *normalized Discounted Cumulative Gain* (nDCG)@10 to evaluate both re-ranking and retrieval performance. For the re-ranking, we restrict to re-ranking only the top 1000 candidate document provided as part of the dataset to be fair, while some works use stronger methods to find better top 1000 candidates for re-ranking (resulting in higher evaluation numbers) See Appendix D for a detailed discussion on these evaluation metrics. Finally, we also evaluate EmbedDistill on the BEIR benchmark [57] in terms of nDCG@10 and recall@100 metrics.

249 **Model architectures.** We follow the standard Transformers-based IR model architectures similar
250 to Karpukhin et al. [20], Qu et al. [48], Oğuz et al. [47]. We utilized various sizes of DE models based
251 on BERT-base [11] (12-layer, 768 dim, 110M parameters), DistilBERT [55] (6-layer, 768 dim, 67.5M
252 parameters – $\sim$ 2/3 of base), or BERT-mini [58] (4-layer, 256 dim, 11.3M parameters – $\sim$ 1/10 of
253 base). For query generation (cf. Sec. 4.3), we employ BART-base [27], an encoder-decoder model, to
254 generate similar questions from each training example's input question (query). We randomly mask
255 10% of tokens and inject zero mean Gaussian noise with $\sigma = \{0.1, 0.2\}$ between the encoder and
256 decoder. See Appendix E for more details on query generation and Appendix F.1 for hyperparameters.

## 5.2 DE to DE distillation

258 We employ AR2 [63][3] and SentenceBERT-
259 v5 [50][4] as teacher DE models for NQ
260 and MSMARCO. Note that both models
261 are based on BERT-base. For DE to DE
262 distillation, we consider two kinds of con-
263 figurations for the student DE model: (1)
264 *Symmetric*: We use identical question and
265 document encoders. We evaluate Distil-
266 BERT and BERT-mini on both datasets. (2)
267 *Asymmetric*: The student inherits document
268 embeddings from the teacher DE model
269 and *are not* trained during the distillation.
270 For query encoder, we use DistilBERT or
271 BERT-mini which are smaller than docu-
272 ment encoder.

273 **Student DE model training.** We train stu-
274 dent DE models using a combination of
275 (i) one-hot loss (cf. Eq. 9 in Appendix A)
276 on training data; (ii) distillation loss in
277 (cf. Eq. 11 in Appendix A); and (iii) em-

Table 3: Performance of various DE models on MSMARCO
dev set for both *re-ranking* and *retrieval* tasks (full corpus).
The teacher model (110.1M parameter BERT-base models
as both encoders) for re-ranking achieves MRR@10 of 36.8
and that for retrieval get MRR@10 of 37.2. The table shows
performance (in MRR@10) of the symmetric DE student
model (67.5M or 11.3M transformer as both encoders), and
asymmetric DE student model (67.5M or 11.3M transformer
as query encoder and document embeddings inherited from
the teacher).

| Method | Re-ranking | | Retrieval | |
|---|---|---|---|---|
| | 67.5M | 11.3M | 67.5M | 11.3M |
| Train student directly | 27.0 | 23.0 | 22.6 | 18.6 |
| + Distill from teacher | 34.6 | 30.4 | 35.0 | 28.6 |
| + Inherit doc embeddings | 35.2 | 32.1 | 35.7 | 30.3 |
| + Query embedding matching | 36.2 | **35.0** | 35.4 | **40.8** |
| + Query generation | 36.2 | 34.4 | 37.2 | 34.8 |
| Train student using only embedding matching and inherit doc embeddings | **36.5** | 33.5 | **36.6** | 31.4 |
| + Query generation | 36.4 | 34.1 | **36.7** | 32.8 |

278 bedding matching loss in Eq. 7. We used `[CLS]`-pooling for all student encoders. Unlike DPR [20]
279 or AR2, we do not use hard negatives from BM25 or other models, which greatly simplifies our
280 distillation procedure.

281 **Results and discussion.** To understand the impact of various proposed configurations and losses, we
282 train models by sequentially adding components and evaluate their retrieval performance on NQ and
283 MSMARCO dev set as shown in Table 1 and Table 3 respectively. (See Table 6 in Appendix F.2 for
284 performance on NQ in terms of the relaxed recall and Table 7 in Appendix F.3 for MSMARCO in
285 terms of nDCG@10.)

286 We begin by training a symmetric DE without distillation. As expected, moving to distillation brings
287 in considerable gains. Next, we swap the student document encoder with document embeddings
288 from the teacher (non-trainable), which leads to a good jump in the performance. Now we can
289 introduce EmbedDistill with Eq. 7 for aligning query representations between student and teacher.
290 The two losses are combined with weight of 1.0 (except for BERT-mini models in the presence of
291 query generation with 5.0). This improves performance significantly, e.g.,it provides $\sim$3 and $\sim$5
292 points increase in recall@5 on NQ with students based on DistilBERT and BERT-mini, respectively
293 (Table 1). We further explore the utility of EmbedDistill in aligning the teacher and student embedding
294 spaces in Appendix G.1.

295 On top of the two losses (standard distillation and embedding matching), we also use $R_{\mathrm{Emb,Q}}(t, s; Q')$
296 from Sec. 4.3 on 2 additional questions (per input question) generated from BART. We also try a
297 variant where we eliminate the standard distillation loss and only employ the embedding matching
298 loss in Eq. 7 along with inheriting teacher's document embeddings. This configuration without the
299 standard distillation loss leads to excellent performance (with query generation again providing
300 additional gains in most cases.)

---

[3] https://github.com/microsoft/AR2/tree/main/AR2
[4] https://huggingface.co/sentence-transformers/msmarco-bert-base-dot-v5

It is worth highlighting that DE models trained with the proposed methods (e.g., asymmetric DE with embedding matching and generation) achieve 99% of the performance in both NQ/MSMARCO tasks with a query encoder that is 2/3rd the size of that of the teacher. Furthermore, even with 1/10th size of the query encoder, our proposal can achieve 95-97% of the performance. This is particularly useful for latency critical applications with minimal impact on the final performance.

Finally, we take our best student models, i.e., one trained using with additional embedding matching loss and using data augmentation from query generation, and evaluate on test sets. We compare with various prior work and note that most prior work used considerably bigger models in terms of parameters,

Table 4: Average BEIR performance of our DE teacher and EmbedDistill student models and their numbers of trainable parameters. Both models are trained on MSMARCO and evaluated on 14 other datasets (the average does not include MS-MARCO). The full table is at Appendix F.4. With EmbedDistill, student materializes most of the performance of the teacher on the unforeseen datasets.

| Method | #Layers | nDCG@10 | R@100 |
|---|---|---|---|
| DPR [21] | 12 | 22.5 | 47.7 |
| ANCE [60] | 12 | 40.5 | 60.0 |
| TAS-B [15] | 6 | 42.8 | 64.8 |
| GenQ [57] | 6 | 42.5 | 64.2 |
| Our teacher [50] | 12 (220.2M) | 45.7 | 65.1 |
| EmbedDistill | 6 (67.5M) | 44.0 | 63.5 |

depth (12 or 24 layers), or width (upto 1024 dims). For NQ test set results are reported in Table 2, but as MSMARCO does not have any public test set, we instead present results for the BEIR benchmark in Table 4. Note we also provide evaluation of our SentenceBERT teacher achieving very high performance on the benchmark which can be of independent interest (please refer to Appendix F.4 for details). For both NQ and BEIR, our approach obtains competitive student model with fewer than 50% of the parameters: even with 6 layers, our student model is very close (98-99%) to its teacher.

### 5.3 CE to DE distillation

We consider two CE teachers for MSMARCO re-ranking task[5]: a standard [CLS]-pooled CE teacher, and the Dual-pooled CE teacher (cf. Sec. 4.2). Both teachers are based on RoBERTa-base and trained on triples in the training set for 300K steps with cross-entropy loss.

**Student DE model training.** We considered the following distillation variants: standard score-based distillation from the [CLS]-pooled teacher, and our novel Dual-pooled CE teacher (with and without embedding matching loss). For each variant, we initialize encoders of the student DE model with two RoBERTa-base models and train for 500K steps on the training triples. We performed the naïve joint embedding

Table 5: Performance of DE models distilled from [CLS]-pooled and Dual-pooled CE models on MS-MARCO re-ranking task (original top1000 dev). While both teacher models perform similarly, embedding matching-based distillation only works with the Dual-pooled teacher. See Appendix F for nDCG@10 metric.

| Method | MRR@10 |
|---|---|
| [CLS]-pooled teacher | 37.1 |
| Dual-pooled teacher | 37.0 |
| Standard distillation from [CLS]-pooled teacher | 33.0 |
| +Joint matching | 32.4 |
| Standard distillation from Dual-pooled teacher | 33.3 |
| +Query matching | **33.7** |

matching for the [CLS]-pooled teacher (cf. Sec. 4.2) and employed the query embedding matching (cf. Eq.7) for the Dual-pooled CE teacher. In either case, embedding-matching loss is added on top of the standard cross entropy loss with the weight of $1.0$ (when used).

**Results and discussion.** Table 5 evaluates the effectiveness of the dual pooling and the embedding matching for CE to DE distillation. As described in Sec. 4.2, the traditional [CLS]-pooled teacher did not provide any useful embedding for the embedding matching (see Appendix G.2 for the further analysis of the resulting embedding space). However, with the Dual-pooled teacher, embedding matching does boost student's performance.

## 6 Related work

Here, we position our EmbedDistill work with respect to prior work on distillation and data augmentation for Transformers-based IR models. We also cover prior efforts on aligning representations during distillation for *non-IR* settings. Unlike our problem setting where the DE student is factorized, these works mainly consider distilling a single large Transformer into a smaller one.

**Distillation for IR.** Traditional distillation techniques have been widely applied in the IR literature, often to distill a teacher CE model to a student DE model [28, 8]. Recently, distillation from a DE

---

[5]Note: Full retrieval is prohibitively expensive with CE models.

model (with complex late interaction) to another DE model (with inner-product scoring) has also been considered [29, 15]. As for distilling across different model architectures, Lu et al. [31], Izacard and Grave [16] consider distillation from a teacher CE model to a student DE model. Hofstätter et al. [14] conduct an extensive study of knowledge distillation across a wide-range of model architectures. Most existing distillation schemes for IR rely on only teacher scores; by contrast, we propose a geometric approach that also utilizes the teacher *embeddings*. Many recent efforts [48, 51, 56] show that iterative multi-stage (self-)distillation improves upon single-stage distillation [48, 51, 56]. These approaches use a model from the previous stage to obtain labels [56] as well as mine harder-negatives [60]. We only focus on the single-stage distillation in this paper. Multi-stage procedures are complementary to our work, as one can employ our proposed embedding-matching approach in various stages of such a procedure. Interestingly, we demonstrate in Sec. 5 that our proposed EmbedDistill can successfully benefit from high quality models trained with such complex procedures [50, 63]. In particular, our single-stage distillation method can transfer almost all of their performance gains to even smaller models. Also to showcase that our method brings gain orthogonal to how teacher was trained, we conduct experiments with single-stage trained teacher in Appendix F.5.

**Distillation with representation alignments.** Outside of the IR context, a few prior works proposed to utilize alignment between hidden layers during distillation [52, 55, 18, 1, 64]. Chen et al. [7] utilize the representation alignment to re-use teacher's classification layer for image classification. Unlike these works, our work is grounded in a rigorous theoretical understanding of the teacher-student (generalization) gap for IR models. Further, our work differs from these as it needs to address multiple challenges presented by an IR setting: 1) cross-architecture distillation such as CE to DE distillation; 2) partial representation alignment of query or document representations as opposed to aligning for the entire input, i.e., a query-documents pair; and 3) catering representation alignment approach to novel IR setups such as asymmetric DE configuration. To the best of our knowledge, our work is first in the IR literature that goes beyond simply matching scores (or its proxies) for distillation.

**Semi-supervised learning for IR.** Data augmentation or semi-supervised learning has been previously used to ensure data efficiency in IR [see, e.g., 35, 66]. More interestingly, data augmentation have enabled performance improvements as well. Doc2query [45, 44] performs document expansion by generating queries that are relevant to the document and appending those queries to the document. Query expansion has also been considered, e.g., for document re-ranking [67]. Notably, generating synthetic (query, passage, answer) triples from a text corpus to augment existing training data for QA systems also leads to significant gains [2, 47]. Furthermore, even zero-shot approaches, where no labeled query-document pairs are used, can also perform competitively to supervised methods [26, 17, 33, 54]. Unlike these works, we utilize query-generation capability to ensure tighter alignment between the embedding spaces of the teacher and student.

**Richer transformers-based architectures for IR.** Besides DE and CE models (cf. Sec. 2), intermediate configurations [36, 22, 42, 32] have been proposed. Such models independently encode query and document before applying a more complex *late interaction* between the two. Nogueira et al. [46] explore *generative* encoder-decoder style model for re-ranking. In this paper, we focus on basic DE/CE models to showcase the benefits of our proposed geometric distillation approach. Exploring embedding matching for aforementioned architectures is an interesting avenue for future work.

## 7 Conclusion

We propose EmbedDistill — a novel distillation method for IR that goes beyond simple score matching. En route, we provide a theoretical understanding of the teacher-student generalization gap in an IR setting which not only motivated EmbedDistill but also inspired new design choices for the student DE models: (a) reusing the teacher's document encoder in the student and (b) aligning query embeddings of the teacher and student. This simple approach delivers consistent quality and computational gains in practical deployments and we demonstrate them on MSMARCO, NQ, and BEIR benchmarks. Finally, we found EmbedDistill retains 95-97% of the teacher performance to with 1/10th size students.

**Limitations.** As discussed in Sec. 4.2 and 5.3, EmbedDistill requires modifications in the CE scoring function to be effective. In terms of underlying IR model architectures, we only explore Transformer-based models in our experiments; primarily due to their widespread utilization. That said, we expect our results to extend to non-Transformer architectures such as MLPs. Finally, we note that our experiments only consider NLP domains, and exploring other modalities (e.g., vision) or multi-modal settings (e.g., image-to-text search) is left as an interesting avenue for future work.

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
