# EmbedDistill: A Geometric Knowledge Distillation for Information Retrieval (Supplementary)

## Correction of Table 3 in the main body

We found that the Table 3 in our manuscript contains a couple of typos (mixing MRR with NDCG) and have corrected with the below (updated entry is in red color). We will update the table in the final version of the paper.

Table 3: Performance of various DE models on MSMARCO dev set for both *re-ranking* and *retrieval* tasks (full corpus). The teacher model (110.1M parameter BERT-base models as both encoders) for re-ranking achieves MRR@10 of 36.8 and that for retrieval get MRR@10 of 37.2. The table shows performance (in MRR@10) of the symmetric DE student model (67.5M or 11.3M transformer as both encoders), and asymmetric DE student model (67.5M or 11.3M transformer as query encoder and document embeddings inherited from the teacher).

| Method | Re-ranking | | Retrieval | |
|---|---|---|---|---|
| | 67.5M | 11.3M | 67.5M | 11.3M |
| Train student directly | 27.0 | 25.0 | 22.6 | 18.6 |
| + Distill from teacher | 34.6 | 30.4 | 35.0 | 28.6 |
| + Inherit doc embeddings | 35.2 | 32.1 | 35.7 | 30.3 |
| + Query embedding matching | 36.2 | **35.1** | 37.1 | **35.4** |
| + Query generation | 36.2 | 34.4 | **37.2** | 34.8 |
| Train student using only embedding matching and inherit doc embeddings | **36.5** | 33.5 | 36.6 | 31.4 |
| + Query generation | 36.5 | 34.1 | 36.7 | 32.8 |

## A  Loss functions

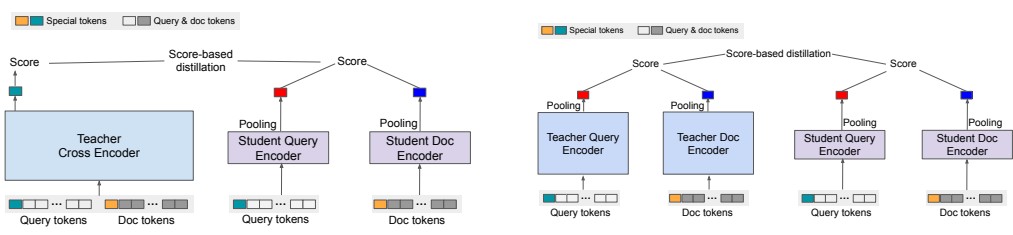

(a) Score-based CE to DE distillation

(b) Score-based DE to DE distillation

Figure 2: Illustration of score-based distillation for IR (cf. Section 2.2). Fig. 2a describes distillation from a teacher [CLS]-pooled CE model to a student DE model. Fig. 2b depicts distillation from a teacher DE model to a student DE model. Here, both distillation setups employ symmetric DE configurations where query and document encoders of the student model are of the same size.

Here, we state various (per-example) loss functions that most commonly define training objectives for IR models. Typically, one hot training with original label is performed using *softmax-based cross-entropy loss* functions:

$$\ell\big(s_{q,\mathbf{d}_i}, \mathbf{y}_i\big) \;=\; -\sum_{j \in [L]} y_{i,j} \cdot \log\Big(\frac{\exp(s(q_i, d_{i,j}))}{\sum\limits_{j' \in [L]} \exp(s(q_i, d_{i,j'}))}\Big). \tag{9}$$

Alternatively, it is also common to employ a one-vs-all loss function based on *binary cross-entropy loss* as follows:

$$\ell\big(s_{q,\mathbf{d}_i}, \mathbf{y}_i\big) \;=\; -\sum_{j \in [L]} \Big(y_{i,j} \cdot \log\Big(\frac{1}{1 \,+\, \exp(-s(q_i, d_{i,j}))}\Big) \;+$$
$$(1 - y_{i,j}) \cdot \log\Big(\frac{1}{1 \,+\, \exp(s(q_i, d_{i,j}))}\Big)\Big). \tag{10}$$

Note that $\mathbf{d}_i = \{d_{i,j}\}_{j \in [L]}$ can be expanded to include various forms of negatives such as in-batch negatives [21] and sampled negatives [3].

As for distillation (cf. Figure 2), one can define a distillation objective based on the softmax-based cross-entropy loss as:[6]

$$\ell_{\mathrm{d}}\big(s^{\mathrm{s}}_{q,\mathbf{d}_i}, s^{\mathrm{t}}_{q,\mathbf{d}_i}\big) = -\sum_{j \in [L]} \left( \frac{\exp(s^{\mathrm{t}}_{i,j})}{\sum_{j' \in [L]} \exp(s^{\mathrm{t}}_{i,j'})} \cdot \log \left( \frac{\exp(s^{\mathrm{s}}_{i,j})}{\sum_{j' \in [L]} \exp(s^{\mathrm{s}}_{i,j'})} \right) \right), \qquad (11)$$

where $s^{\mathrm{t}}_{i,j} := s^{\mathrm{t}}(q_i, d_{i,j})$ and $s^{\mathrm{s}}_{i,j} := s^{\mathrm{s}}(q_i, d_{i,j})$ denote the teacher and student scores, respectively. On the other hand, the distillation objective with the binary cross-entropy takes the form:

$$\ell_{\mathrm{d}}\big(s^{\mathrm{s}}_{q,\mathbf{d}_i}, s^{\mathrm{t}}_{q,\mathbf{d}_i}\big) = -\sum_{j \in [L]} \left( \frac{1}{1 + \exp(-s^{\mathrm{t}}_{i,j})} \cdot \log \left( \frac{1}{1 + \exp(-s^{\mathrm{s}}_{i,j})} \right) + \right.$$
$$\left. \frac{1}{1 + \exp(s^{\mathrm{t}}_{i,j})} \cdot \log \left( \frac{1}{1 + \exp(s^{\mathrm{s}}_{i,j})} \right) \right). \qquad (12)$$

Finally, distillation based on the meas square error (MSE) loss (aka. logit matching) employs the following loss function:

$$\ell_{\mathrm{d}}\big(s^{\mathrm{s}}_{q,\mathbf{d}_i}, s^{\mathrm{t}}_{q,\mathbf{d}_i}\big) = \sum_{j \in [L]} \big( s^{\mathrm{t}}(q_i, d_{i,j}) - s^{\mathrm{s}}(q_i, d_{i,j}) \big)^2. \qquad (13)$$

# B   Dual pooling details

In this work, we focus on two kinds of dual pooling strategies:

- **Special tokens-based dual pooling.** Let $\mathrm{pool}_{\mathrm{CLS}}$ and $\mathrm{pool}_{\mathrm{SEP}}$ denote the pooling operations that return the embeddings of the [CLS] and [SEP] tokens, respectively. We define

$$\mathrm{emb}^{\mathrm{t}}_{q \leftarrow (q,d)} = \mathrm{pool}_{\mathrm{CLS}}\big(\mathrm{Enc}^{\mathrm{t}}(\tilde{o})\big),$$
$$\mathrm{emb}^{\mathrm{t}}_{d \leftarrow (q,d)} = \mathrm{pool}_{\mathrm{SEP}}\big(\mathrm{Enc}^{\mathrm{t}}(\tilde{o})\big), \qquad (14)$$

  where $\tilde{o}$ denotes the input token sequence to the Transformers-based encoder, which consists of { query, document, special } tokens.

- **Segment-based weighted-mean dual pooling.** Let $\mathrm{Enc}^{\mathrm{t}}(\tilde{o})|_Q$ and $\mathrm{Enc}^{\mathrm{t}}(\tilde{o})|_D$ denote the final query token embeddings and document token embeddings produced by the encoder, respectively. We define the *proxy* query and document embeddings

$$\mathrm{emb}^{\mathrm{t}}_{q \leftarrow (q,d)} = \mathrm{mean}_{\mathrm{wt}}\big(\mathrm{Enc}^{\mathrm{t}}(\tilde{o})|_Q\big),$$
$$\mathrm{emb}^{\mathrm{t}}_{d \leftarrow (q,d)} = \mathrm{mean}_{\mathrm{wt}}\big(\mathrm{Enc}^{\mathrm{t}}(\tilde{o})|_D\big), \qquad (15)$$

  where $\mathrm{mean}_{\mathrm{wt}}(\cdot)$ denotes the weighted mean operation. We employ the specific weighting scheme where each token receives a weight equal to the inverse of the square root of the token-sequence length.

# C   Deferred details and proofs from Section 3

In this section we present more precise statements and proofs of Theorem 3.1 and Proposition 3.2 (stated informally in Section 3 of the main text) along with the necessary background. First, for the ease of exposition, we define new notation which will facilitate theoretical analysis in this section.

**Notation.** Denote the query and document encoders as $f \colon \mathcal{Q} \to \mathbb{R}^k$ and $g \colon \mathcal{D} \to \mathbb{R}^k$ for the student, and $F \colon \mathcal{Q} \to \mathbb{R}^k, G \colon \mathcal{D} \to \mathbb{R}^k$ for the teacher (in the dual-encoder setting). With $q$ denoting a query and $d$ denoting a document, $f(q)$ and $g(d)$ then denote query and document embeddings, respectively, generated by the student. We define $F(q)$ and $G(d)$ similarly for embeddings by the teacher.[7]

---

[6]It is common to employ temperature scaling with softmax operation. We do not explicitly show the temperature parameter for ease of exposition.

[7]Note that, as per the notations in the main text, we have $(f, g) = (\mathrm{Enc}^{\mathrm{s}}_Q, \mathrm{Enc}^{\mathrm{s}}_D)$ and $(F, G) = (\mathrm{Enc}^{\mathrm{t}}_Q, \mathrm{Enc}^{\mathrm{t}}_D)$. Similarly, we have $(\mathrm{emb}^{\mathrm{t}}_q, \mathrm{emb}^{\mathrm{t}}_d) = (f(q), g(d))$ and $(\mathrm{emb}^{\mathrm{t}}_q, \mathrm{emb}^{\mathrm{t}}_d) = (F(q), G(d))$.

**Theorem C.1** (Formal statement of Theorem 3.1). *Let $\mathcal{F}$ and $\mathcal{G}$ denote the function classes for the query and document encoders for the student model, respectively. Given $n$ examples $\mathcal{S}_n = \{(q_i, d_i, y_i)\}_{i\in[n]} \subset \mathcal{Q} \times \mathcal{D} \times \{0,1\}$, let $s^{\mathrm{s}}(q,d) := s^{f,g}(q_i, d_i) = f(q_i)^T g(d_i)$ be the scores assigned to the $(q_i, d_i)$ pair by a dual-encoder model with $f \in \mathcal{F}$ and $g \in \mathcal{G}$ as query and document encoders, respectively. Let $\ell$ and $\ell_{\mathrm{d}}$ be the binary cross-entropy loss (cf. Eq. 10 with $L = 1$) and the distillation-specific loss based on it (cf. Eq. 12 with $L = 1$), respectively. In particular,*

$$\ell(s^{F,G}(q_i, d_i), y_i) := -y_i \log \sigma\left(F(q_i)^\top G(d_i)\right) - (1 - y_i) \log\left[1 - \sigma\left(F(q_i)^\top G(d_i)\right)\right]$$

$$\ell_{\mathrm{d}}(s^{f,g}(q_i, d_i), s^{F,G}(q_i, d_i)) := -\sigma\left(F(q_i)^\top G(d_i)\right) \cdot \log \sigma\left(f(q_i)^\top g(d_i)\right) - \left[1 - \sigma\left(F(q_i)^\top G(d_i)\right)\right] \cdot \log\left[1 - \sigma\left(f(q_i)^\top g(d_i)\right)\right],$$

*where $\sigma$ is the sigmoid function and $s^{\mathrm{t}} := s^{F,G}$ denotes the teacher dual-encoder model with $F$ and $G$ as its query and document encoders, respectively. Assume that*

1. *All encoders $f, g, F,$ and $G$ have the same output dimension.*

2. *$\exists\ K \in (0,\infty)$ such that $\sup_{q\in\mathcal{Q}} \max\left\{\|f(q)\|_2, \|F(q)\|_2\right\} \leq K$ and $\sup_{d\in\mathcal{D}} \max\left\{\|g(d)\|_2, \|G(d)\|_2\right\} \leq K$.*

*Then, we have*

$$\underbrace{\mathbb{E}\left[s^{f,g}(q,d)\right]}_{:=R(s^{\mathrm{s}})=R(s^{f,g})} - \underbrace{\mathbb{E}\left[s^{F,G}(q,d)\right]}_{:=R(s^{\mathrm{t}})=R(s^{F,G})} \leq \underbrace{\sup_{(f,g)\in\mathcal{F}\times\mathcal{G}} \left|R(s^{f,g}, s^{F,G}; \mathcal{S}_n) - \mathbb{E}\left[\ell_{\mathrm{d}}\left(s^{f,g}(q,d), s^{F,G}(q,d)\right)\right]\right|}_{:=\mathcal{E}_n(\mathcal{F},\mathcal{G})}$$

$$+ 2K\Big(\underbrace{\frac{1}{n}\sum_{i\in[n]} \|g(d_i) - G(d_i)\|_2}_{:=R_{\mathrm{Emb},D}(\mathrm{t},\mathrm{s};\mathcal{S}_n)} + \underbrace{\frac{1}{n}\sum_{i\in[n]} \|f(q_i) - F(q_i)\|_2}_{:=R_{\mathrm{Emb},Q}(\mathrm{t},\mathrm{s};\mathcal{S}_n)}\Big) + \underbrace{R(s^{F,G}; \mathcal{S}_n) - R(s^{F,G})}_{:=\Delta(s^{\mathrm{t}};\mathcal{S}_n)}$$

$$+ K^2\Big(\mathbb{E}\left[\left|\sigma(F(q)^\top G(d)) - y\right|\right] + \frac{1}{n}\sum_{i\in[n]} \left|\sigma\left(F(q_i)^\top G(d_i)\right) - y_i\right|\Big). \tag{16}$$

*Proof.* Note that

$$R(s^{f,g}) - R(s^{F,G}) = R(s^{f,g}) - R(s^{f,g}, s^{F,G}) + R(s^{f,g}, s^{F,G}) - R(s^{F,G})$$

$$\overset{(a)}{\leq} K^2\mathbb{E}\left[\left|\sigma(F(q)^\top G(d)) - y\right|\right] + R(s^{f,g}, s^{F,G}) - R(s^{F,G})$$

$$= K^2\mathbb{E}\left[\left|\sigma(F(q)^\top G(d)) - y\right|\right] + R(s^{f,g}, s^{F,G}) - R(s^{f,g}, s^{F,G}; \mathcal{S}_n) + R(s^{f,g}, s^{F,G}; \mathcal{S}_n) - R(s^{F,G})$$

$$\overset{(b)}{\leq} K^2\mathbb{E}\left[\left|\sigma(F(q)^\top G(d)) - y\right|\right] + \mathcal{E}_n(\mathcal{F},\mathcal{G}) + R(s^{f,g}, s^{F,G}; \mathcal{S}_n) - R(s^{F,G})$$

$$= K^2\mathbb{E}\left[\left|\sigma(F(q)^\top G(d)) - y\right|\right] + \mathcal{E}_n(\mathcal{F},\mathcal{G}) + R(s^{f,g}, s^{F,G}; \mathcal{S}_n) - R(s^{F,G}; \mathcal{S}_n) + R(s^{F,G}; \mathcal{S}_n) - R(s^{F,G})$$

$$\overset{(c)}{\leq} K^2\mathbb{E}\left[\left|\sigma(F(q)^\top G(d)) - y\right|\right] + \mathcal{E}_n(\mathcal{F},\mathcal{G}) + \underbrace{R(s^{F,G}; \mathcal{S}_n) - R(s^{F,G})}_{:=\Delta(s^{\mathrm{t}};\mathcal{S}_n)} +$$

$$\frac{2K}{n}\sum_{i\in[n]} \|g(d_i) - G(d_i)\|_2 + \frac{2K}{n}\sum_{i\in[n]} \|f(q_i) - F(q_i)\|_2 +$$

$$\frac{K^2}{n}\sum_{i\in[n]} \left|\sigma\left(F(q_i)^\top G(d_i)\right) - y_i\right| \tag{17}$$

where $(a)$ follows from Lemma C.3, $(b)$ follows from the definition of $\mathcal{E}_n(\mathcal{F},\mathcal{G})$, and $(c)$ follows from Proposition C.2. $\qquad\square$

### C.1 Bounding the difference between student's empirical *distillation* risk and teacher's empirical risk

**Lemma C.2.** *Given $n$ examples $\mathcal{S}_n = \{(q_i, d_i, y_i)\}_{i \in [n]} \subset \mathcal{Q} \times \mathcal{D} \times \{0, 1\}$, let $s^{f,g}(q_i, d_i) = f(q_i)^T g(d_i)$ be the scores assigned to the $(q_i, d_i)$ pair by a dual-encoder model with $f$ and $g$ as query and document encoders, respectively. Let $\ell$ and $\ell_{\mathrm{d}}$ be the binary cross-entropy loss (cf. Eq. 10 with $L = 1$) and the distillation-specific loss based on it (cf. Eq. 12 with $L = 1$), respectively. In particular,*

$$\ell(s^{F,G}(q_i, d_i), y_i) := -y_i \log \sigma \left( F(q_i)^\top G(d_i) \right) - (1 - y_i) \log \left[ 1 - \sigma \left( F(q_i)^\top G(d_i) \right) \right]$$

$$\ell_{\mathrm{d}}(s^{f,g}(q_i, d_i), s^{F,G}(q_i, d_i)) := -\sigma \left( F(q_i)^\top G(d_i) \right) \cdot \log \sigma \left( f(q_i)^\top g(d_i) \right) -$$
$$\left[ 1 - \sigma \left( F(q_i)^\top G(d_i) \right) \right] \cdot \log \left[ 1 - \sigma \left( f(q_i)^\top g(d_i) \right) \right],$$

*where $\sigma$ is the sigmoid function and $s^{F,G}$ denotes the teacher dual-encoder model with $F$ and $Q$ as its query and document encoders, respectively. Assume that*

1. *All encoders $f, g, F$, and $G$ have the same output dimension $k \geq 1$.*

2. *$\exists \ K \ \in \ (0, \infty)$ such that $\sup_{q \in \mathcal{Q}} \max \{ \|f(q)\|_2, \|F(q)\|_2 \} \leq K$ and $\sup_{d \in \mathcal{D}} \max \{ \|g(d)\|_2, \|G(d)\|_2 \} \leq K$.*

*Then, we have*

$$\frac{1}{n} \sum_{i \in [n]} \ell_{\mathrm{d}} \left( s^{f,g}(q_i, d_i), s^{F,G}(q_i, d_i) \right) - \frac{1}{n} \sum_{i \in [n]} \ell \left( s^{F,G}(q_i, d_i), y_i \right) \leq$$
$$\frac{2K}{n} \sum_{i \in [n]} \|g(d_i) - G(d_i)\|_2 + \frac{2K}{n} \sum_{i \in [n]} \|f(q_i) - F(q_i)\|_2 +$$
$$\frac{K^2}{n} \sum_{i \in [n]} \left| \sigma \left( F(q_i)^\top G(d_i) \right) - y_i \right|. \tag{18}$$

*Proof.* We first note that the distillation loss can be rewritten as

$$\ell_{\mathrm{d}} \left( s^{f,g}(q, d), s^{F,G}(q, d) \right) = \left( 1 - \sigma(F(q)^\top G(d)) \right) f(q)^\top g(d) + \gamma(-f(q)^\top g(d)),$$

where $\gamma(v) := \log[1 + e^v]$ is the softplus function. Similarly, the one-hot (label-dependent) loss can be rewritten as

$$\ell \left( s^{F,G}(q, d), y \right) = (1 - y) F(q)^\top G(d) + \gamma(-F(q)^\top G(d)).$$

Recall from our notation in Section 2 that

$$R(s^{f,g}, s^{F,G}; \mathcal{S}_n) := \frac{1}{n} \sum_{i \in [n]} \ell_{\mathrm{d}} \left( s^{f,g}(q_i, d_i), s^{F,G}(q_i, d_i) \right), \tag{19}$$

$$R(s^{F,G}; \mathcal{S}_n) := \frac{1}{n} \sum_{i \in [n]} \ell \left( s^{F,G}(q_i, d_i), y_i \right), \tag{20}$$

as the empirical risk based on the distillation loss, and the empirical risk based on the label-dependent loss, respectively. With this notation, the quantity to upper bound can be rewritten as

$$R(s^{f,g}, s^{F,G}; \mathcal{S}_n) - R(s^{F,G}; \mathcal{S}_n) = \underbrace{R(s^{f,g}, , s^{F,G}; \mathcal{S}_n) - R(s^{f,G}, s^{F,G}; \mathcal{S}_n)}_{:= \square_1} +$$
$$\underbrace{R(s^{f,G}, s^{F,G}; \mathcal{S}_n) - R(s^{F,G}, s^{F,G}; \mathcal{S}_n)}_{:= \square_2} + \underbrace{R(s^{F,G}, s^{F,G}; \mathcal{S}_n) - R(s^{F,G}; \mathcal{S}_n)}_{:= \square_3}. \tag{21}$$

We start by bounding $\square_1$ as

$$\square_1 = \frac{1}{n} \sum_{i \in [n]} \left( \ell_{\mathrm{d}} \left( s^{f,g}(q_i, d_i), s^{F,G}(q_i, d_i) \right) - \ell_{\mathrm{d}} \left( s^{f,G}(q_i, d_i), s^{F,G}(q_i, d_i) \right) \right)$$

$$= \frac{1}{n} \sum_{i \in [n]} \Big( \big(1 - \sigma(F(q_i)^\top G(d_i))\big) f(q_i)^\top g(d_i) + \gamma(-f(q_i)^\top g(d_i))$$

$$- \big(1 - \sigma(F(q_i)^\top G(d_i))\big) f(q_i)^\top G(d_i) - \gamma(-f(q_i)^\top G(d_i)) \Big)$$

$$= \frac{1}{n} \sum_{i \in [n]} \Big( f(q_i)^\top \big(g(d_i) - G(d_i)\big) \big(1 - \sigma(F(q_i)^\top G(d_i))\big)$$

$$+ \gamma(-f(q_i)^\top g(d_i)) - \gamma(-f(q_i)^\top G(d_i)) \Big)$$

$$\overset{(a)}{\leq} \frac{1}{n} \sum_{i \in [n]} \Big( f(q_i)^\top \big(g(d_i) - G(d_i)\big) \big(1 - \sigma(F(q_i)^\top G(d_i))\big) + \big| f(q_i)^\top g(d_i) - f(q_i)^\top G(d_i) \big| \Big)$$

$$\overset{(b)}{\leq} \frac{1}{n} \sum_{i \in [n]} \Big( \|f(q_i)\| \|g(d_i) - G(d_i)\| \big(1 - \sigma(F(q_i)^\top G(d_i))\big) + \|f(q_i)\| \|g(d_i) - G(d_i)\| \Big)$$

$$\leq \frac{K}{n} \sum_{i \in [n]} \|g(d_i) - G(d_i)\|_2 \big(2 - \sigma(F(q_i)^\top G(d_i))\big) \Big)$$

$$\leq \frac{2K}{n} \sum_{i \in [n]} \|g(d_i) - G(d_i)\|_2, \tag{22}$$

where at $(a)$ we use the fact that $\gamma$ is a Lipschitz continuous function with Lipschitz constant 1, and at $(b)$ we use Cauchy-Schwarz inequality.

Similarly for $\square_2$, we proceed as

$$\square_2 = \frac{1}{n} \sum_{i \in [n]} \Big( \ell_d\big(s^{f,G}(q_i, d_i), s^{F,G}(q_i, d_i)\big) - \ell_d\big(s^{F,G}(q_i, d_i), s^{F,G}(q_i, d_i)\big) \Big)$$

$$= \frac{1}{n} \sum_{i \in [n]} \Big( \big(1 - \sigma(F(q_i)^\top G(d_i))\big) f(q_i)^\top G(d_i) + \gamma(-f(q_i)^\top G(d_i))$$

$$- \big(1 - \sigma(F(q_i)^\top G(d_i))\big) F(q_i)^\top G(d_i) - \gamma(-F(q_i)^\top G(d_i)) \Big)$$

$$= \frac{1}{n} \sum_{i \in [n]} \Big( G(d_i)^\top (f(q_i) - F(q_i)) \big(1 - \sigma(F(q_i)^\top G(d_i))\big)$$

$$+ \gamma(-f(q_i)^\top G(d_i)) - \gamma(-F(q_i)^\top G(d_i)) \Big)$$

$$\leq \frac{1}{n} \sum_{i \in [n]} \Big( \|G(d_i)\| \|f(q_i) - F(q_i)\| + \big| f(q_i)^\top G(d_i) - F(q_i)^\top G(d_i) \big| \Big)$$

$$\leq \frac{2K}{n} \sum_{i \in [n]} \|f(q_i) - F(q_i)\|_2. \tag{23}$$

$\square_3$ can be bounded as

$$\square_3 = R(s^{F,G}, s^{F,G}; \mathcal{S}_n) - R(s^{F,G}; \mathcal{S}_n)$$

$$= \frac{1}{n} \sum_{i \in [n]} \Big( \ell_d\big(s^{F,G}(q_i, d_i), s^{F,G}(q_i, d_i)\big) - \ell\big(s^{F,G}(q_i, d_i), y_i\big) \Big)$$

$$= \frac{1}{n} \sum_{i \in [n]} \Big( \big(1 - \sigma(F(q_i)^\top G(d_i))\big) F(q_i)^\top G(d_i) + \gamma(-F(q_i)^\top G(d_i))$$

$$- (1 - y_i) F(q_i)^\top G(d_i) - \gamma(-F(q_i)^\top G(d_i)) \Big)$$

$$= \frac{1}{n} \sum_{i \in [n]} \Big( \big(1 - \sigma(F(q_i)^\top G(d_i)) - (1 - y_i)\big) F(q_i)^\top G(d_i) \Big)$$

$$\leq \frac{K^2}{n} \sum_{i \in [n]} \left| \sigma(F(q_i)^\top G(d_i)) - y_i \right|. \tag{24}$$

Combining Eq. 21, 22, 23, and 24 establishes the bound in Eq. 18. $\qquad \square$

**Lemma C.3.** *Given an example $(q, d, y) \in \mathcal{Q} \times \mathcal{D} \times \{0, 1\}$, let $s^{f,g}(q, d) = f(q)^T g(d)$ be the scores assigned to the $(q, d)$ pair by a dual-encoder model with $f$ and $g$ as query and document encoders, respectively. Let $\ell$ and $\ell_{\mathrm{d}}$ be the binary cross-entropy loss (cf. Eq. 10 with $L = 1$) and the distillation-specific loss based on it (cf. Eq. 12 with $L = 1$), respectively. In particular,*

$$\ell(s^{f,g}(q, d), y) := -y \log \sigma \left( f(q)^\top g(d) \right) - (1 - y) \log \left[ 1 - \sigma \left( f(q)^\top g(d) \right) \right]$$

$$\ell_{\mathrm{d}}(s^{f,g}(q, d), s^{F,G}(q, d)) := -\sigma \left( F(q)^\top G(d) \right) \cdot \log \sigma \left( f(q)^\top g(d) \right) -$$
$$\left[ 1 - \sigma \left( F(q)^\top G(d) \right) \right] \cdot \log \left[ 1 - \sigma \left( f(q)^\top g(d) \right) \right],$$

*where $\sigma$ is the sigmoid function and $s^{F,G}$ denotes the teacher dual-encoder model with $F$ and $Q$ as its query and document encoders, respectively. Assume that*

1. *All encoders $f, g, F$, and $G$ have the same output dimension $k \geq 1$.*

2. *$\exists \ K \in (0, \infty)$ such that $\sup_{q \in \mathcal{Q}} \max \{\|f(q)\|_2, \|F(q)\|_2\} \leq K$ and $\sup_{d \in \mathcal{D}} \max \{\|g(d)\|_2, \|G(d)\|_2\} \leq K$.*

*Then, we have*

$$\underbrace{\mathbb{E} \left[ \ell \left( s^{f,g}(q, d), y \right) \right]}_{:= R(s^{f,g})} - \underbrace{\mathbb{E} \left[ \ell_{\mathrm{d}} \left( s^{f,g}(q, d), s^{F,G}(q, d) \right) \right]}_{:= R(s^{f,g}, s^{F,G})} \leq K_Q K_D \mathbb{E} \left[ \left| \sigma(F(q)^\top G(d)) - y \right| \right] \tag{25}$$

*where expectation are defined by a joint distribution $\mathbb{P}(q, d, y)$ over $\mathcal{Q} \times \mathcal{D} \times \{0, 1\}$*

*Proof.* Similar to the proof of Proposition C.2, we utilize the fact that

$$\ell \left( s^{F,G}(q, d), y \right) = (1 - y) F(q)^\top G(d) + \gamma(-F(q)^\top G(d)),$$
$$\ell_{\mathrm{d}} \left( s^{f,g}(q, d), s^{F,G}(q, d) \right) = \left( 1 - \sigma(F(q)^\top G(d)) \right) f(q)^\top g(d) + \gamma(-f(q)^\top g(d)),$$

where $\gamma(v) := \log[1 + e^v]$ is the softplus function. Now,

$$\mathbb{E} \left[ \ell \left( s^{f,g}(q, d), y \right) - \ell_{\mathrm{d}} \left( s^{f,g}(q, d), s^{F,G}(q, d) \right) \right] \tag{26}$$
$$= \mathbb{E} \left[ (1 - y) f(q)^\top g(d) + \gamma(-f(q)^\top g(d)) \right]$$
$$\quad - \mathbb{E} \left[ \left( 1 - \sigma(F(q)^\top G(d)) \right) f(q)^\top g(d) + \gamma(-f(q)^\top g(d)) \right]$$
$$= \mathbb{E} \left[ \left( 1 - y - \left( 1 - \sigma(F(q)^\top G(d)) \right) \right) F(q)^\top G(d) \right]$$
$$\leq K^2 \mathbb{E} \left[ \left| \sigma(F(q)^\top G(d)) - y \right| \right], \tag{27}$$

which completes the proof. $\qquad \square$

## C.2 Uniform deviation bound

Let $\mathcal{F}$ denote the class of functions that map queries in $\mathcal{Q}$ to their embeddings in $\mathbb{R}^k$ via the query encoder. Define $\mathcal{G}$ analogously for the doc encoder, which consists of functions that map documents in $\mathcal{D}$ to their embeddings in $\mathbb{R}^k$. To simplify exposition, we assume that each training example consists of a single relevant or irrelevant document for each query, i.e., $L = 1$ in Section 2. Let

$$\mathcal{F}\mathcal{G} = \{(q, d) \mapsto f(q)^\top g(d) \ | \ f \in \mathcal{F}, g \in \mathcal{G}\}$$

Given $\mathcal{S}_n = \{(q_i, d_i, y_i) : i \in [n]\}$, let $N(\epsilon, \mathcal{H})$ denote the $\epsilon$-covering number of a function class $\mathcal{H}$ with respect to $L_2(\mathbb{P}_n)$ norm, where $\|h\|_{L_2(\mathbb{P}_n)}^2 := \|h\|_n^2 := \frac{1}{n} \sum_{i=1}^n \|h(q_i, d_i)\|_2^2$. Depending on the context, the functions in $\mathcal{H}$ may map to $\mathbb{R}$ or $\mathbb{R}^d$.

**Proposition C.4.** *Let $s^{\mathrm{t}}$ be scorer of a teacher model and $\ell_{\mathrm{d}}$ be a distillation loss function which is $L_{\ell_{\mathrm{d}}}$-Lipschitz in its first argument. Let the embedding functions in $\mathcal{F}$ and $\mathcal{G}$ output vectors with $\ell_2$ norms at most $K$. Define the uniform deviation*

$$\mathcal{E}_n(\mathcal{F}, \mathcal{G}) = \sup_{f \in \mathcal{F}, g \in \mathcal{G}} \left| \frac{1}{n} \sum_{i \in [n]} \ell_{\mathrm{d}}\big(f(q_i)^\top g(d_i), s^{\mathrm{t}}_{q_i, d_i}\big) - \mathbb{E}_{q,d} \ell_{\mathrm{d}}\big(f(q)^\top g(d), s^{\mathrm{t}}_{q,d}\big) \right|.$$

*For any $g^* \in \mathcal{G}$, we have*

$$\mathbb{E}_{\mathcal{S}_n} \mathcal{E}_n(\mathcal{F}, \mathcal{G}) \leq \mathbb{E}_{\mathcal{S}_n} \frac{48 K L_{\ell_{\mathrm{d}}}}{\sqrt{n}} \int_0^\infty \sqrt{\log N(u, \mathcal{F}) + \log N(u, \mathcal{G})} \, du,$$

$$\mathbb{E}_{\mathcal{S}_n} \mathcal{E}_n(\mathcal{F}, \{g^*\}) \leq \mathbb{E}_{\mathcal{S}_n} \frac{48 K L_{\ell_{\mathrm{d}}}}{\sqrt{n}} \int_0^\infty \sqrt{\log N(u, \mathcal{F})} \, du.$$

*Proof of Proposition C.4.* We first symmetrize excess risk to get Rademacher complexity, then bound the Rademacher complexity with Dudley's entropy integral.

For a training set $\mathcal{S}_n$, the empirical Rademacher complexity of a class of functions $\mathcal{H}$ that maps $\mathcal{Q} \times \mathcal{D}$ to $\mathbb{R}$ is defined by

$$\mathrm{Rad}_n(\mathcal{H}) = \mathbb{E}_\sigma \sup_{h \in \mathcal{H}} \frac{1}{n} \sum_{i=1}^n \varepsilon_i h(q_i, d_i),$$

where $\{\varepsilon_i\}$ denote i.i.d. Rademacher random variables taking the value in $\{+1, -1\}$ with equal probability. By symmetrization [4] and the fact that $\ell_{\mathrm{d}}$ is $L_{\ell_{\mathrm{d}}}$-Lipschitz in its first argument, we get

$$E_{\mathcal{S}_n} \mathcal{E}_n(\mathcal{F}, \mathcal{G}) \leq 2 L_{\ell_{\mathrm{d}}} \mathbb{E}_{\mathcal{S}_n} \mathrm{Rad}_n(\mathcal{F}\mathcal{G}).$$

Then, Dudley's entropy integral [see, e.g., 25] gives

$$\mathrm{Rad}_n(\mathcal{F}\mathcal{G}) \leq \frac{12}{\sqrt{n}} \int_0^\infty \sqrt{\log N(u, \mathcal{F}\mathcal{G})} \, du.$$

From Lemma C.5 with $K_Q = K_D = K$, for any $u > 0$,

$$N(u, \mathcal{F}\mathcal{G}) \leq N\left(\frac{u}{2K}, \mathcal{F}\right) N\left(\frac{u}{2K}, \mathcal{G}\right).$$

Putting these together,

$$\mathbb{E}_{\mathcal{S}_n} \mathcal{E}_n(\mathcal{F}, \mathcal{G}) \leq \frac{24 L_{\ell_{\mathrm{d}}}}{\sqrt{n}} \int_0^\infty \sqrt{\log N(u/2K, \mathcal{F}) + \log N(u/2K, \mathcal{G})} \, du. \tag{28}$$

Following the same steps with $\mathcal{G}$ replaced by $\{g^*\}$, we get

$$\mathbb{E}_{\mathcal{S}_n} \mathcal{E}_n(\mathcal{F}, \{g^*\}) \leq \frac{24 L_{\ell_{\mathrm{d}}}}{\sqrt{n}} \int_0^\infty \sqrt{\log N(u/2K, \mathcal{F})} \, du \tag{29}$$

By changing variable in Eq. 28 and Eq. 29, we get the stated bounds. □

For $f : \mathcal{Q} \to \mathbb{R}^k, g : \mathcal{D} \to \mathbb{R}^k$, define $fg : \mathcal{Q} \times \mathcal{D} \to \mathbb{R}$ by $fg(q, d) = f(q)^\top g(d)$.

**Lemma C.5.** *Let $f_1, \ldots, f_N$ be an $\epsilon$-cover of $\mathcal{F}$ and $g_1, \ldots, g_M$ be an $\epsilon$-cover of $\mathcal{G}$ in $L_2(\mathbb{P}_n)$ norm. Let $\sup_{f \in \mathcal{F}} \sup_{q \in \mathcal{Q}} \|f(q)\|_2 \leq K_Q$ and $\sup_{g \in \mathcal{G}} \sup_{d \in \mathcal{D}} \|g(d)\|_2 \leq K_D$. Then,*

$$\{f_i g_j \mid i \in [N], j \in [M]\}$$

*is a $(K_Q + K_D)\epsilon$-cover of $\mathcal{F}\mathcal{G}$.*

*Proof of Lemma C.5.* For arbitrary $f \in \mathcal{F}, g \in \mathcal{G}$, there exist $\tilde{f} \in \{f_1, \ldots, f_N\}, \tilde{g} \in \{g_1, \ldots, g_M\}$ such that $\|f - \tilde{f}\|_n \leq \epsilon, \|g - \tilde{g}\|_n \leq \epsilon$. It is sufficient to show that $\|fg - \tilde{f}\tilde{g}\|_n \leq (K_Q + K_D)\epsilon$. Decomposing using triangle inequality,

$$\|fg - \tilde{f}\tilde{g}\|_n = \|fg - f\tilde{g} + f\tilde{g} - \tilde{f}\tilde{g}\|_n$$
$$\leq \|fg - f\tilde{g}\|_n + \|f\tilde{g} - \tilde{f}\tilde{g}\|_n. \tag{30}$$

To bound the first term, using Cauchy-Schwartz inequality, we can write

$$\frac{1}{n}\sum_{i=1}^{n}\left(f(q_i)^\top g(d_i) - \tilde{f}(q_i)^\top \tilde{g}(d_i)\right)^2 \leq \sup_{q\in\mathcal{Q}}\|f(q)\|_2^2 \cdot \frac{1}{n}\sum_{i=1}^{n}\|(g-\tilde{g})(d_i)\|_2^2.$$

Therefore

$$\|fg - f\tilde{g}\|_n \leq K_Q\|g - \tilde{g}\|_n \leq K_Q\epsilon.$$

Similarly

$$\|f\tilde{g} - \tilde{f}\tilde{g}\|_n \leq K_D\|f - \tilde{f}\|_n \leq K_D\epsilon$$

Plugging these in Eq. 30, we get

$$\|fg - \tilde{f}\tilde{g}\|_n \leq (K_Q + K_D)\epsilon.$$

This completes the proof. ☐

# D Evaluation metric details

For NQ, we evaluate models with full *strict* recall metric, meaning that the model is required to find a *golden* passage from the whole set of candidates (21M). Specifically, for $k \geq 1$, recall@$k$ or R@$k$ denotes the percentage of questions for which the associated golden passage is among the $k$ passages that receive the highest relevance scores by the model. In addition, we also present results for *relaxed* recall metric considered by Karpukhin et al. [20], where R@$k$ denotes the percentage of questions where the corresponding answer string is present in at least one of the $k$ passages with the highest model (relevance) scores.

For both MSMARCO retrieval and re-ranking tasks, we follow the standard evaluation metrics *Mean Reciprocal Rank*(MRR)@10 and *normalized Discounted Cumulative Gain* (nDCG)@10. For retrieval tasks, these metrics are computed with respect to the whole set of candidates passages (8.8M). On the other hand, for re-ranking task, the metrics are computed with respect to BM25 generated 1000 candidate passages –*the originally provided*– for each query. Please note that some papers use more powerful models (e.g., DE models) to generate the top 1000 candidate passages, which is not a standard re-ranking evaluation and should not be compared directly. We report $100 \times$ MRR@10 and $100 \times$ nDCG@10, as per the convention followed in the prior works.

# E Query generation details

We introduced query generation to encourage geometric matching in local regions, which can aid in transferring more knowledge in confusing neighborhoods. As expected, this further improves the distillation effectiveness on top of the embedding matching in most cases. To focus on the local regions, we generate queries from the observed examples by adding local perturbation in the data manifold (embedding space). Specifically, we employ an off-the-shelf encoder-decoder model – BART-base [27]. First, we embed an observed query in the corresponding dataset. Second, we add a small perturbation to the query embedding. Finally, we decode the perturbed embedding to generate a new query in the input space. Formally, the generated query $x'$ given an original query $x$ takes the form $x' = \text{Dec}(\text{Enc}(x) + \epsilon)$, where $\text{Enc}()$ and $\text{Dec}()$ correspond to the encoder and the decoder from the off-the-shelf model, respectively, and $\epsilon$ is an isotropic Gaussian noise. Furthermore, we also randomly mask the original query tokens with a small probability. We generate two new queries from an observed query and use them as additional data points during our distillation procedure.

As a comparison, we tried adding the same size of random sampled queries instead of the ones generated via the method described above. That did not show any benefit, which justifies the use of our query/question generation method.

# F Experimental details and additional results

## F.1 Additional training details

**Optimization.** For all of our experiments, we use ADAM weight decay optimizer with a short warm up period and a linear decay schedule. We use the initial learning rate of $10^{-5}$ and $2.8 \times 10^{-5}$ for experiments on NQ and MSMARCO, respectively. We chose batch sizes to be 128.

## F.2 Additional results on NQ

See Table 6 for the performance of various DE models on NQ, as measured by the *relaxed* recall metric.

Table 6: *Relaxed* recall performance of various student DE models on NQ dev set, including symmetric DE student model (67.5M or 11.3M transformer for both encoders), and asymmetric DE student model (67.5M or 11.3M transformer as query encoder and document embeddings inherited from the teacher). All distilled students used the same teacher (110M parameter BERT-base models as both encoders), with the performance (in terms of relaxed recall) of Recall@5 = 87.2, Recall@20 = 94.7, Recall@100 = 98.1. *Note: the proposed method can achieve 100% of teacher's performance even with $2/3^{rd}$ size of the query encoder, and 92-97% with even $1/10^{th}$ size.*

| Method | Recall@5 | | Recall@20 | | Recall@100 | |
|---|---|---|---|---|---|---|
| | 67.5M | 11.3M | 67.5M | 11.3M | 67.5M | 11.3M |
| Train student directly | 62.5 | 49.7 | 82.5 | 73.0 | 93.7 | 88.2 |
| + Distill from teacher | 82.7 | 66.1 | 92.9 | 84.0 | 97.3 | 93.1 |
| + Inherit document embeddings | 84.7 | 73.0 | 93.7 | 85.4 | 97.6 | 93.3 |
| + Query embedding matching | 87.2 | 77.6 | **95.0** | 88.0 | 97.9 | 94.3 |
| + Query generation | **87.8** | **80.3** | 94.8 | **89.9** | **98.0** | **95.6** |
| Train student only using embedding matching and inherit doc embeddings | 86.4 | 69.1 | 94.2 | 81.6 | 97.7 | 89.9 |
| + Query generation | 86.7 | 72.9 | 94.4 | 84.9 | 97.8 | 92.2 |

## F.3 Additional results on MSMARCO

### F.3.1 DE to DE distillation

See Table 7 for DE to DE distillation results on MSMARCO retrieval and re-ranking task, as measured by the nDCG@10 metric (see Section 5.2 for the results on MRR@10 metric).

Table 7: Performance of various DE models on MSMARCO dev set for both *re-ranking* (original top1000) and *retrieval* tasks (full corpus). The teacher model (110.1M parameter BERT-base models as both encoders) for reranking achieves nDCG@10 of 42.7 and that for retrieval get nDCG@10 44.2. The table shows performance (in nDCG@10) of the symmetric DE student model (67.5M or 11.3M transformer as both encoders), and asymmetric DE student model (67.5M or 11.3M transformer as query encoder and document embeddings inherited from the teacher).

| Method | Re-ranking | | Retrieval | |
|---|---|---|---|---|
| | 67.5M | 11.3M | 67.5M | 11.3M |
| Train student directly | 32.2 | 29.7 | 27.2 | 22.5 |
| + Distill from teacher | 40.2 | 35.8 | 41.3 | 34.1 |
| + Inherit doc embeddings | 41.0 | 37.7 | 42.2 | 36.2 |
| + Query embedding matching | 42.0 | **40.8** | **43.8** | **41.9** |
| + Query generation | 42.0 | 40.1 | **43.8** | 41.2 |
| Train student using only embedding matching and inherit doc embeddings | **42.3** | 39.3 | 43.3 | 37.6 |
| + Query generation | **42.3** | 39.9 | 43.4 | 39.2 |

### F.3.2 CE to DE distillation

See Table 8 for CE to DE distillation results on MSMARCO re-ranking task, as measured by the nDCG@10 metric (see Section 5.3 for the results on MRR@10 metric).

**CE to DE distillation with stronger teacher model.** Recall that the CE to DE distillation exploration in Section 5.3 employs a dual-pooled RoBERTa-base model as the teacher. We now utilize a much stronger CE teacher model to further showcase the effectiveness of EmbedDistill for CE to DE distillation on MSMARCO re-ranking task. In particular, we convert SimLM [CLS]-pooled CE model[8] to a dual-pooled CE model via standard score-based distillation (cf. Section 2.2). We subsequently utilize the resulting dual-pooled version of the SimLM CE model as a teacher to perform CE to DE

---

[8]https://github.com/microsoft/unilm/tree/master/simlm

Table 8: Performance of CE to DE distillation on MSMARCO re-ranking task, as measured by the nDCG@10 metric. As for the teacher CE models, we consider two kinds of CE models based on two different pooling mechanism.

| Method | nDCG@10 |
|---|---|
| `[CLS]`-pooled teacher | 43.0 |
| Dual-pooled teacher | 42.8 |
| Standard distillation from `[CLS]`-teacher | 38.8 |
|   +Joint matching | 38.0 |
| Standard distillation from Dual-pooling teacher | 39.2 |
|   +Query matching | **39.4** |

distillation via embedding alignment. Similar to DE to DE distillation (cf. Section 5.2), we aim to identify the utility of various components of EmbedDistill in our exploration. See Table 9 for the results.

Interestingly, we also explore distilling dual-pooled CE model to an asymmetric DE model. In this setting, DE model simply inherits the document embeddings from the CE model. Crucially, the inheritance of the document embedding from the dual-pooled CE model can be done offline as we feed an *empty query* along with the document (separated by the `[SEP]` token) to obtain the document embedding from the dual-pooled CE model. Thus, the excellent performance of distillation to an asymmetric DE model (which inherits document embeddings from the dual-pooled CE model) not only showcases the power of embedding alignment via EmbedDistill but it also highlights the effectiveness of dual-pooling method employed at the teacher.

Table 9: Performance of various DE models obtained via CE to DE distillation on MSMARCO dev set for *re-ranking* (original top1000). The teacher model is a dual-pooled version of the SimLM model which achieves MRR@10 of 40.0 nDCG@10 of 45.8. The table shows performance of the symmetric DE student model (67.5M as both encoders), and asymmetric DE student model (67.5M transformer as query encoder and document embeddings inherited from the dual-pooled teacher). Note that the document embeddings used during inheritance are generated in a query-independent manner from the CE teacher model (with an *empty* query).

| Method | MRR@10 | nDCG@10 |
|---|---|---|
| Train student directly | 27.0 | 32.2 |
|   + Distill from teacher | 33.2 | 38.7 |
|   + Inherit doc embeddings | 35.4 | 41.0 |
|   + Query embedding matching | 36.1 | 41.7 |
|   + Query generation | 36.3 | 42.0 |
| Train student using only embedding matching and inherit doc embeddings | **36.9** | **42.6** |
|   + Query generation | 36.8 | 42.5 |
| Standard distillation from `[CLS]`-teacher | 32.8 | 38.4 |

## F.4 Additional results on BEIR benchmark

See Table 10 (NDCG@10) and Table 11 (Recall@100) for BEIR benchmark results. All numbers are from BEIR benchmark paper [57]. As common practice, non-public benchmark sets[9], {BioASQ, Signal-1M(RT), TREC-NEWS, Robust04}, are removed from the table. Following the original BEIR paper [57] (Table 9 and Appendix G from the original paper), we utilized Capped Recall@100 for TREC-COVID dataset.

---

[9]https://github.com/beir-cellar/beir

Table 10: In-domain and zero-shot retrieval performance on BEIR benchmark [57], as measured by **nDCG@10**. All the baseline number in the table are taken from [57]. We exclude (in-domain) MSMARCO from average computation as common practice.

| Model (→) | Lexical | Sparse | | | Dense | | | | | |
|---|---|---|---|---|---|---|---|---|---|---|
| Dataset (↓) | **BM25** | **DeepCT** | **SPARTA** | **docT5query** | **DPR** | **ANCE** | **TAS-B** | **GenQ** | **SentenceBERT** *(our teacher)* | **EmbedDistill** *(ours)* |
| MS MARCO | 22.8 | 29.6‡ | 35.1‡ | 33.8‡ | 17.7 | 38.8‡ | 40.8‡ | 40.8‡ | 47.1‡ | 46.6‡ |
| TREC-COVID | 65.6 | 40.6 | 53.8 | 71.3 | 33.2 | 65.4 | 48.1 | 61.9 | 75.4 | 72.3 |
| NFCorpus | 32.5 | 28.3 | 30.1 | 32.8 | 18.9 | 23.7 | 31.9 | 31.9 | 31.0 | 30.7 |
| NQ | 32.9 | 18.8 | 39.8 | 39.9 | 47.4‡ | 44.6 | 46.3 | 35.8 | 51.5 | 50.8 |
| HotpotQA | 60.3 | 50.3 | 49.2 | 58.0 | 39.1 | 45.6 | 58.4 | 53.4 | 58.0 | 56.0 |
| FiQA-2018 | 23.6 | 19.1 | 19.8 | 29.1 | 11.2 | 29.5 | 30.0 | 30.8 | 31.8 | 29.5 |
| ArguAna | 31.5 | 30.9 | 27.9 | 34.9 | 17.5 | 42.9 | 42.9 | 49.3 | 38.5 | 34.9 |
| Touché-2020 | 36.7 | 15.6 | 17.5 | 34.7 | 13.1 | 24.0 | 16.2 | 18.2 | 22.9 | 24.7 |
| CQADupStack | 29.9 | 26.8 | 25.7 | 32.5 | 15.3 | 29.6 | 31.4 | 34.7 | 33.5 | 30.6 |
| Quora | 78.9 | 69.1 | 63.0 | 80.2 | 24.8 | 85.2 | 83.5 | 83.0 | 84.2 | 81.4 |
| DBPedia | 31.3 | 17.7 | 31.4 | 33.1 | 26.3 | 28.1 | 38.4 | 32.8 | 37.7 | 35.9 |
| SCIDOCS | 15.8 | 12.4 | 12.6 | 16.2 | 07.7 | 12.2 | 14.9 | 14.3 | 14.8 | 14.4 |
| FEVER | 75.3 | 35.3 | 59.6 | 71.4 | 56.2 | 66.9 | 70.0 | 66.9 | 76.7 | 76.9 |
| Climate-FEVER | 21.3 | 06.6 | 08.2 | 20.1 | 14.8 | 19.8 | 22.8 | 17.5 | 23.5 | 22.5 |
| SciFact | 66.5 | 63.0 | 58.2 | 67.5 | 31.8 | 50.7 | 64.3 | 64.4 | 59.8 | 55.5 |
| AVG (w/o MSMARCO) | 43.0 | 31.0 | 35.5 | 44.4 | 25.5 | 40.5 | 42.8 | 42.5 | 45.7 | 44.0 |

Table 11: In-domain and zero-shot retrieval performance on BEIR benchmark [57], as measured by **Recall@100**. All the baseline number in the table are taken from [57]. ‡ indicates in-domain retrieval performance. ∗ indicates capped recall following original benchmark setup. We exclude (in-domain) MSMARCO from average computation as common practice.

| Model (→) | Lexical | Sparse | | | Dense | | | | | |
|---|---|---|---|---|---|---|---|---|---|---|
| Dataset (↓) | **BM25** | **DeepCT** | **SPARTA** | **docT5query** | **DPR** | **ANCE** | **TAS-B** | **GenQ** | **SentenceBERT** *(our teacher)* | **EmbedDistill** *(ours)* |
| MS MARCO | 65.8 | 75.2‡ | 79.3‡ | 81.9‡ | 55.2 | 85.2‡ | 88.4‡ | 88.4‡ | 91.7‡ | 90.6‡ |
| TREC-COVID | 49.8* | 34.7* | 40.9* | 54.1* | 21.2* | 45.7* | 38.7* | 45.6* | 54.1* | 48.8* |
| NFCorpus | 25.0 | 23.5 | 24.3 | 25.3 | 20.8 | 23.2 | 28.0 | 28.0 | 27.7 | 26.7 |
| NQ | 76.0 | 63.6 | 78.7 | 83.2 | 88.0‡ | 83.6 | 90.3 | 86.2 | 91.1 | 89.9 |
| HotpotQA | 74.0 | 73.1 | 65.1 | 70.9 | 59.1 | 57.8 | 72.8 | 67.3 | 69.7 | 68.3 |
| FiQA-2018 | 53.9 | 48.9 | 44.6 | 59.8 | 34.2 | 58.1 | 59.3 | 61.8 | 62.0 | 60.1 |
| ArguAna | 94.2 | 93.2 | 89.3 | 97.2 | 75.1 | 93.7 | 94.2 | 97.8 | 89.2 | 87.8 |
| Touché-2020 | 53.8 | 40.6 | 38.1 | 55.7 | 30.1 | 45.8 | 43.1 | 45.1 | 45.3 | 45.5 |
| CQADupStack | 60.6 | 54.5 | 52.1 | 63.8 | 40.3 | 57.9 | 62.2 | 65.4 | 63.9 | 61.3 |
| Quora | 97.3 | 95.4 | 89.6 | 98.2 | 47.0 | 98.7 | 98.6 | 98.8 | 98.5 | 98.1 |
| DBPedia | 39.8 | 37.2 | 41.1 | 36.5 | 34.9 | 31.9 | 49.9 | 43.1 | 46.0 | 42.6 |
| SCIDOCS | 35.6 | 31.4 | 29.7 | 36.0 | 21.9 | 26.9 | 33.5 | 33.2 | 32.5 | 31.5 |
| FEVER | 93.1 | 73.5 | 84.3 | 91.6 | 84.0 | 90.0 | 93.7 | 92.8 | 93.9 | 93.8 |
| Climate-FEVER | 43.6 | 23.2 | 22.7 | 42.7 | 39.0 | 44.5 | 53.4 | 45.0 | 49.3 | 47.6 |
| SciFact | 90.8 | 89.3 | 86.3 | 91.4 | 72.7 | 81.6 | 89.1 | 89.3 | 88.9 | 87.2 |
| AVG (w/o MSMARCO) | 63.4 | 55.9 | 56.2 | 64.7 | 47.7 | 60.0 | 64.8 | 64.2 | 65.1 | 63.5 |

## F.5 Additional results with single-stage trained teachers

Hereby we evaluate EmbedDistill with a simple single-stage trained teachers instead of teachers trained in complex multi-stage frameworks, in order to test the generalizability of the method.

Similar to Table 1, we conducted an experiment on top of single-stage trained teacher based on RoBERTa-base instead of AR2 [63] in the main text. We also changed the student to be based on DistilRoBERTa or RoBERTa-mini accordingly for simplicity to use same tokenizer.

Table 12 demonstrates that EmbedDistill provides a significant boost of the performance on top of standard distillation techniques similar to what we observed in Table 1.

Table 12: *Full* recall performance of various student DE models on NQ dev set, including symmetric DE student model, and asymmetric DE student models. All students used the same *in-house teacher* (124M parameter RoBERTa-base models as both encoders), with the full Recall@5 = 64.6, Recall@20 = 81.7, and Recall@100 = 91.5.

| Method | 6-Layer (82M) | | | 4-Layer (16M) | | |
|---|---|---|---|---|---|---|
| | R@5 | R@20 | R@100 | R@5 | R@20 | R@100 |
| Train student directly | 41.9 | 64.5 | 82.0 | 39.5 | 59.9 | 76.3 |
| + Distill from teacher | 48.3 | 67.2 | 80.9 | 44.9 | 61.1 | 74.8 |
| + Inherit doc embeddings | 56.9 | 74.3 | 85.4 | 47.2 | 64.0 | 77.0 |
| + Query embedding matching | 61.8 | 78.7 | 89.0 | 56.7 | 74.6 | 85.9 |
| + Query generation | 61.7 | 79.4 | 89.6 | 57.1 | 75.2 | **86.7** |
| Train student using only embedding matching and inherit doc embeddings | 63.7 | 80.3 | 90.3 | 57.9 | 74.6 | 85.7 |
| + Query generation | **64.1** | **80.5** | **90.4** | **58.9** | **76.0** | 86.6 |

Furthermore, we also consider a in-house trained teacher (RoBERTa-base) for MSMARCO re-ranking task. Table 13 demonstrates a similar pattern to Table 3, providing evidence of generalizability of EmbedDistill.

Table 13: Reranking performance of various DE models on MSMARCO dev set. We utilize a RoBERTa-base in-house trained teacher achieving MRR@10 of 33.1 and nDCG@10 of 38.8 is used. The table shows performance of the symmetric DE student model and asymmetric DE student models.

| Method | MRR@10 | | nDCG@10 | |
|---|---|---|---|---|
| | 82M | 16M | 82M | 16M |
| Train student directly | 29.7 | 26.3 | 35.2 | 31.4 |
| + Distill from teacher | 31.6 | 28.4 | 37.2 | 33.5 |
| + Inherit doc embeddings | 32.4 | 30.2 | 38.0 | 35.8 |
| + Query embedding matching | 32.8 | 31.9 | 38.6 | 37.6 |
| + Query generation | 33.0 | **32.0** | 38.8 | **37.7** |
| Train student only using embedding matching and inherit doc embeddings | 32.7 | 31.8 | 38.5 | 37.5 |
| + Query generation | **33.0** | 31.8 | **38.9** | 37.5 |

These result showcase that our method brings performance boost orthogonal to how teacher was trained, whether single-staged or multi-staged.

## G  Embedding analysis

### G.1  DE to DE distillation

Traditional score matching-based distillation might not result in transfer of relative geometry from teacher to student. To assess this, we look at the discrepancy between the teacher and student query embeddings for all $q, q'$ pairs: $\|\mathtt{emb}_q^t - \mathtt{emb}_{q'}^t\| - \|\mathtt{emb}_q^s - \mathtt{emb}_{q'}^s\|$. Note that the analysis is based on NQ, and we focus on the teacher and student DE models based on BERT-base and DistilBERT, respectively. As evident from Fig. 3, embedding matching loss significantly reduces this discrepancy.

### G.2  CE to DE distillation

We qualitatively look at embeddings from CE model in Fig. 4. The embedding $\mathtt{emb}_{q,d}^t$ from [CLS]-pooled CE model does not capture semantic similarity between query and document as it is solely trained to classify whether the query-document pair is relevant or not. In contrast, the (proxy) query embeddings $\mathtt{emb}_{q \leftarrow (q,d)}^t$ from our Dual-pooled CE model with reconstruction loss do not degenerate and its embeddings groups same query whether conditioned on positive or negative document together. Furthermore, other related queries are closer than unrelated queries. Such informative embedding space would aid distillation to a DE model via embedding matching.

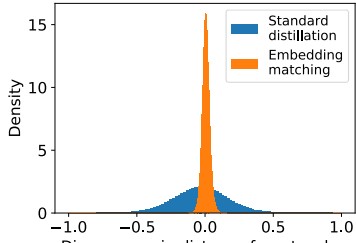

Figure 3: Histogram of teacher-student distance discrepancy in queries.

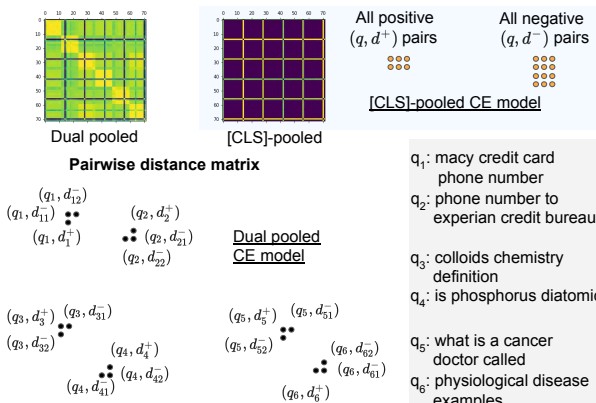

Figure 4: Illustration of geometry expressed by [CLS]-pooled CE and our Dual-pooled CE model on 6 queries from MSMARCO and 12 passages based on pairwise distance matrix across these 72 pairs. [CLS]-pooled CE embeddings degenerates as all positive and negative query-document pairs almost collapse to two points and fail to capture semantic information. In contrast, our Dual-pooled CE model leads to much richer representation that can express semantic information.