# OpenReview forum: "EmbedDistill: A Geometric Knowledge Distillation for Information Retrieval"
_NeurIPS.cc/2023/Conference — Submitted to NeurIPS 2023_

### Official Review · Reviewer_pFfq · 2023-07-04

**Soundness:** 3 good
**Presentation:** 2 fair
**Contribution:** 1 poor
**Rating:** 3
**Confidence:** 5

**Summary:**

This paper presents a distillation technique called EmbedDistill for Information Retrieval, aiming to align the pooled representations of various IR models. The focus of this study revolves around two distillation paradigms: DE-to-DE distillation and CE-to-DE distillation. The former involves straightforward embedding matching, while the latter utilizes special tokens for alignment.

**Strengths:**

1. Embedding matching is a practical method for distillation. This work pioneers the application of feature mapping to the distillation of Information Retrieval (IR) models, contributing to the existing body of knowledge in this area.
2. The experimental findings presented in this study reveal certain positive outcomes when applying EmbedDistill to IR models.


**Weaknesses:**

1. The concept of embedding matching lacks novelty and has been extensively explored in prior studies [1, 2, 3]. Pooling techniques are also commonly employed to handle the matching of intermediate features.
2. The theoretical proof, while complex, does not significantly contribute to understanding the proposed method, as it is essentially a standard feature mapping approach already described in previous works.
3. There appears to be no direct correlation between the term "Geometric" and the proposed method. If a straightforward matching technique can be considered a geometric algorithm, it implies that most feature-based KD methods could also be classified as geometric.

[1] Jiao, Xiaoqi, et al. "Tinybert: Distilling bert for natural language understanding." arXiv preprint arXiv:1909.10351 (2019).
[2] Romero, Adriana, et al. "Fitnets: Hints for thin deep nets." arXiv preprint arXiv:1412.6550 (2014).
[3] Hofstätter, Sebastian, et al. "Improving efficient neural ranking models with cross-architecture knowledge distillation." arXiv preprint arXiv:2010.02666 (2020).


**Questions:**

My main concern lies in the novelty of this work. Please refer to the weaknesses.

---

> ### Author Rebuttal · Authors · 2023-08-07
>
> Thank you for taking the time to review our submission. Please find our point-by-point response to your comments/concerns below.
>
> > The concept of embedding matching lacks novelty and has been extensively explored in prior studies [1, 2, 3]. Pooling techniques are also commonly employed to handle the matching of intermediate features.
>
> Please note that we have provided a detailed discussion of prior work on **score-based** distillation for IR setting in Section 6. In the same section, we also acknowledge  prior work on **distillation with representation (embedding) alignments** for **non-IR** setups. However, we critically note that (to the best of our knowledge) we are the first ones to employ such embedding alignment-based distillation in an IR setting.
>
> [1] and [2] do not focus on IR settings. In particular, [1] explores embedding alignment to compress a **general-purpose** transformer model. As mentioned in our submission, distillation for IR applications presents unique challenges around **cross-architecture distillation**, dealing with negative sampling, partial alignment, etc. For example, in our work, we explore DE-to-DE distillation while inheriting teacher document encoder and CE-to-DE distillation aided by dual pooling. Such opportunities/challenges are not necessarily present in other domains where distillation is employed. Furthermore, we also accompany our proposed embedding-based distillation for IR with a rigorous theoretical analysis/justification, which to the best of our knowledge is also novel.
>
> As for [3], to the best of our understanding, Hofstätter et al. only explore **score-based** distillation for IR models in [3].
>
> [1] Jiao, Xiaoqi, et al. "Tinybert: Distilling bert for natural language understanding." arXiv preprint arXiv:1909.10351 (2019).
>
> [2] Romero, Adriana, et al. "Fitnets: Hints for thin deep nets." arXiv preprint arXiv:1412.6550 (2014).
>
> [3] Hofstätter, Sebastian, et al. "Improving efficient neural ranking models with cross-architecture knowledge distillation." arXiv preprint arXiv:2010.02666 (2020).
>
> > The theoretical proof, while complex, does not significantly contribute to understanding the proposed method, as it is essentially a standard feature mapping approach already described in previous works.
>
> Please note that our theoretical analysis begins by understanding the teacher-student generalization gap in an IR setting, which subsequently justifies the embedding alignment-based distillation. Furthermore, our analysis also inspires novel asymmetric DE configurations where we simply inherit the document encoder of the teacher model without increasing the inference time latency for the student DE model (cf. lines 141 – 146 and lines 157 – 165). Thus, we believe that our theoretical analysis plays an important role in inspiring the overall EmbedDistill framework and rigorously justifying our various design choices.
>
> We do agree that proofs are somewhat lengthy at times, but we have placed all those details in the Appendix to not hinder the readability of the main body.
>
> > There appears to be no direct correlation between the term "Geometric" and the proposed method. If a straightforward matching technique can be considered a geometric algorithm, it implies that most feature-based KD methods could also be classified as geometric.
>
> Thank you for your comment. Indeed, we refer to our proposed method as a “geometric” method as it deals with embedding spaces of the teacher and student models’ query and document representations and aims to better align those spaces by focusing on the distances (similarity) among embeddings for queries and documents. The reviewer is correct in pointing out that other feature-based KD methods would also qualify as “geometric” methods. However, as discussed in an earlier response, the prior work do not deal with “geometric” methods for **IR settings**.
>
> We would be happy to hear reviewer’s thoughts on what techniques/methods they associate with the term “geometric method”.

---

> > ### Comment · Reviewer_pFfq · 2023-08-16
> >
> > Thank you for the feedback.
> >
> > Certain aspects of my concerns (Weakness 2) have been acknowledged and tackled within the response. However, I remain worried about the novelty of the embedding-matching approach, given that it's a general technique applicable to **ALL tasks**. In this regard, I'm inclined to respectfully disagree with the assertion that, "representation matching + X," wherein X={IR, NER, Entity linking, ...}, could be characterized as a novel idea, even as the first attempt within these specific domains.
> >
> > So, I tend to keep my initial score.

---

> > > ### Author Response · Authors · 2023-08-17
> > >
> > > We would like to emphasize that our contribution is not about *blindly* applying embedding-alignment based distillation to an IR setting. In this work, we started by analyzing the performance gap between the teacher and student IR models and found the gap is bounded by the misalignment between the query and document representations (*novel analysis*). This motivated us to develop embedding-matching based solutions as well as **asymmetric DE** architecture since it realizes perfect alignment for the document embedding spaces between the teacher and student models without increasing runtime latency (novel formulation and novel architecture for IR). Moreover, we developed **a novel mechanism** for extraction of query and document representations from the CE model (*dual pooling*) that enables CE to DE distillation via embedding alignment.
> > >
> > > Hence, we would like the reviewer to reconsider their opinion. Even though the work’s algorithms look similar (e.g. alignment itself), it was motivated differently and is solving a different problem with a different formulation.

---

> > > > ### Comment · Reviewer_pFfq · 2023-08-21
> > > >
> > > > Thanks for the detailed comment.
> > > >
> > > > > Our contribution is not about blindly applying embedding-alignment based distillation to an IR.
> > > >
> > > > I agree with the author's perspective that within this study, alignment isn't simply imposed upon Information Retrieval (IR). But actually for all tasks, we need to do some adaptations to make alignment work accordingly. **My opinion is that adapting a very classic algorithm to a new setting is not novel, which can not reach the bar of NeurIPS.**  It's not surprising that alignment works for IR, EL, NER, etc.
> > > >
> > > > > Moreover, we developed a novel mechanism for the extraction of query and document representations from the CE model (dual pooling) that enables CE to DE distillation via embedding alignment.
> > > >
> > > > Pooling is a very simple and general solution for heterogeneous KD [1,2]. For example, Attention KD is a very early work in this field, which just pools the channel dimensions to facilitate KD [1]. I don't think the novelty of this paper should lie in general KD techniques.
> > > >
> > > > Conclusion: This paper should pay more attention to IR, rather than adapting several KD techniques from other fields and show their effectiveness. Besides, the theoretical part is not strongly connected to the IR task. My concern about the novelty is not addressed.
> > > >
> > > > [1] Zagoruyko, Sergey, and Nikos Komodakis. "Paying more attention to attention: Improving the performance of convolutional neural networks via attention transfer." arXiv preprint arXiv:1612.03928 (2016).
> > > > [2] Gao, Lei, and Hui Gao. "Feature Decoupled Knowledge Distillation via Spatial Pyramid Pooling." Proceedings of the Asian Conference on Computer Vision. 2022.

---

> > > > > ### Author Response · Authors · 2023-08-21
> > > > >
> > > > > Thank you again for taking the time to engage with us during the discussion phase and providing a detailed response.
> > > > >
> > > > > While we feel that some of the opinions expressed by the reviewer are subjective, we would like to respond to other specific points raised by them.
> > > > >
> > > > > > Pooling is a very simple and general solution for heterogeneous KD [1,2]. For example, Attention KD is a very early work in this field, which just pools the channel dimensions to facilitate KD [1]. I don't think the novelty of this paper should lie in general KD techniques.
> > > > >
> > > > > > [1] Zagoruyko, Sergey, and Nikos Komodakis. "Paying more attention to attention: Improving the performance of convolutional neural networks via attention transfer." arXiv preprint arXiv:1612.03928 (2016).
> > > > >
> > > > > > [2] Gao, Lei, and Hui Gao. "Feature Decoupled Knowledge Distillation via Spatial Pyramid Pooling." Proceedings of the Asian Conference on Computer Vision. 2022.
> > > > >
> > > > > Thank you for bringing up [1,2]. We would like to point out that both of these works are from the computer vision/image classification domain and differ from our motivation for utilizing **dual pooling** – to extract out meaningful **separate** representations for queries and documents from a cross-encoder (CE) model which are otherwise not available in a vanilla CE model (**see Appendix G for the embedding analysis**). Please note that successful utilization of dual pooling also required modifying the training procedure for the CE model (**see Line 214 -- 217**).
> > > > >
> > > > > > Besides, the theoretical part is not strongly connected to the IR task.
> > > > >
> > > > > Please note that we specifically focus on an overall function class that involves two encoders and the scoring function is defined by the dot-product between the embeddings produced by the two encoders, which is precisely capturing the dual-encoder architecture used in IR (cf. Theorem 3.1). Similarly, our uniform deviation analysis (Proposition 3.2) is again specifically focusing on this dual-encoder architecture.
> > > > >
> > > > > Thus, we do not agree with the conclusion that the theoretical part is not strongly connected to the IR task. If the reviewer feels that our analysis can also be utilized/adapted to study distillation in other domains (which we tend to agree with), that should not be counted as a negative for our contribution.
> > > > >
> > > > >
> > > > > ### Studying distillation for IR even when distillation is a successful technique in other domains
> > > > >
> > > > > As a more general comment, we would also like to bring the reviewers’ attention to the growing literature on even **score-based distillation** in IR setting (see, e.g., [1] -- [5] which have appeared in venues like ICML, ICLR, CVPR and NAACL), despite knowledge distillation being extensively explored in the ML literature. This alludes to the fact that **distillation for IR** presents its unique challenges and addressing those requires novel contributions. Given this, we strongly believe that, if accepted, our work (which, to the best of our knowledge, is the first comprehensive treatment of embedding-alignment based distillation for IR) will be a valuable addition to NeurIPS.
> > > > >
> > > > > [1] Izacard et al. Distilling knowledge from reader to retriever for question answering. In ICLR 2021.
> > > > >
> > > > > [2] Menon et al. In defense of dual-encoders for neural ranking. ICML 2022.
> > > > >
> > > > > [3] Qu et al. RocketQA: An optimized training approach to dense passage retrieval for open-domain question answering. NAACL 2021.
> > > > >
> > > > > [4] Santhanam et al. ColBERTv2: Effective and Efficient Retrieval via Lightweight Late Interaction. NAACL 2022.
> > > > >
> > > > > [5] Miech et al. Thinking Fast and Slow: Efficient Text-to-Visual Retrieval with Transformers. CVPR 2021.

---

### Official Review · Reviewer_QGH5 · 2023-07-04

**Soundness:** 2 fair
**Presentation:** 2 fair
**Contribution:** 2 fair
**Rating:** 5
**Confidence:** 4

**Summary:**

This paper proposes EmbedDistill, a novel distillation approach for IR that directly aligns the embeddings from the student and teacher. The authors conduct a theoretical analysis of the teacher-student generalization gap, strengthening the importance of embedding alignment and suggesting the use of asymmetric encoders. The architecture can be directly applied in DE to DE distillation. For CE to DE distillation, the authors propose dual-pooling to obtain document and query embeddings from CE.

**Strengths:**



- The authors show theoretically that embedding matching can close the teacher-student generalization gap in IR setting, and support the use of asymmetric conﬁgurations.

**Weaknesses:**



- The authors claim that EmbedDistill can be used for CE to DE distillation, but the empirical results show poor performance. It only improves the score distillation method from 33.3 to 33.7 (+0.4), while the teacher score is 37.0. This raises doubts about the effectiveness of EmbedDistill in CE to DE distillation. EmbedDistill appears to only work in DE to DE distillation, which is an easier scenario where embedding matching is trivial. This significantly weakens the contribution of this work, as EmbedDistill can’t learn from CE, the stronger architecture.
    - One possible reason for this is that the embeddings extracted from a dual-pooled teacher may not be good. Since the query and document embeddings are concatenated in the re-ranker, the model can simply make the representations of the two close to each other when the query and document are relevant, without properly distributing the representations as required in the DE retrieval scenario. This poses a major challenge for embedding matching in CE to DE distillation, and the authors do not seem to address it adequately.
- There are errors and inconsistencies in the formulas presented in the paper and the appendix. For instance, in line 130, $R\left(s^{\mathrm{s}}, s^{\mathrm{t}} ; \mathcal{S}_n\right)$ is not defined (should it be the empirical risk?). In the proof of Lemma C.3, the authors use $K_Q$ and $K_D$ in Eq. (25) but use $K$ elsewhere. The last line of Eq. (26) is incorrect; it should be $=\mathbb{E}\left[\left(1-y-\left(1-\sigma\left(F(q)^{\top} G(d)\right)\right)\right) f(q)^{\top} g(d)\right]$. The authors should thoroughly check all formulas and correct any errors.
- The document needs improvement in terms of writing quality.
    - Section 5, which covers the result analysis, is difficult to follow.
        - The authors use too many parentheses to add additional explanations, which interrupts the reading flow.
        - The captions of the tables contain too much information.
        - The teacher performance is presented in the content of the table of NQ test and BEIR, but is described in the caption of the table of NQ dev and MS MARCO dev, which can confuse the reader.
    - The author uses too much italics in the introduction, which obscures the key points of the paper.

**Questions:**

Please explain Figure 4 in Appendix G.

**Limitations:**

As pointed out in the Weaknesses, EmbedDistill does not seem to work in CE to DE distillation, which limits the contribution.

---

> ### Author Rebuttal · Authors · 2023-08-08
>
> Thank you for your thorough review and detailed comment.
>
> > The authors claim that EmbedDistill can be used for CE to DE distillation, but the empirical results show poor performance…
>
> For the CE to DE distillation, Table 5 in the main text employs a RoBERTa-based model as the CE teacher. This indeed shows poorer performance compared to the DE to DE distillation results in our submission.
>
> That said, we would like to bring Appendix F.3.2 (Table 9) to the reviewer’s attention, where we utilized a much stronger CE teacher obtained from SimLM [1], which we were able to obtain between the deadline for the main text and appendix. We will include this new result in the **main body** of the final version of the paper.
>
> [1] Wang, Liang, et al. Simlm: Pre-training with representation bottleneck for dense passage retrieval. ACL 2023
>
> With this stronger teacher CE model (MRR@10 of 40.0 nDCG@10 of 45.8), we are able to obtain significantly improved performance for CE to DE distillation via EmbedDistill. In particular, Table 9 shows the utility of four key techniques towards CE to DE distillation: 1) dual pooling; 2) embedding alignment; 3) inheriting document embeddings from the CE model (as enabled by the dual pooling); and 4) query generation. In the revised version, we will include Table 9 in the main text.
>
> Here we copy Table 9 from Appendix F.3.2 for reviewer’s attention.
>
> | Method                                       |  MRR@10  |  nDCG@10 |
> |:---------------------------------------------|---------:|---------:|
> |						    |          |          |
> |Train student directly                        | 27.0     | 32.2     |
> |+ Distill from teacher                        | 33.2     | 38.7     |
> |+ Inherit doc embeddings                      | 35.4     | 41.0     |
> |+ Query embedding matching                    | 36.1     | 41.7     |
> |+ Query generation                            | 36.3     | 42.0     |
> |Only embedding matching and inherit doc embed | **36.9** | **42.6** |
> |+ Query generation                            | 36.8     | 42.5     |
> |                                              |          |          |
> |Standard distillation from [CLS]-teacher      | 32.8     | 38.4     |
>
> Please see Appendix F.3.2 Table 9 for details.
>
> **Given that poor CE to DE distillation in the main text was the main limitation highlighted by the reviewer, in light of the above response with stronger CE to DE distillation results, we respectfully request the reviewer to reconsider their score for our submission.**
>
> > There are errors and inconsistencies in the formulas presented in the paper and the appendix. For instance, in line 130, $R(s^{\rm s}, s^{\rm t}; \mathcal{S}_n)$ ...
>
> Note that $R(s^{\rm s}, s^{\rm t}; \mathcal{S}_n)$ denotes the empirical distillation loss. Please refer to Eq. (3) in the main text.
>
> > In the proof of Lemma C.3, the authors use $K_Q$ and $K_D$ in Eq. (25) but use $K$ elsewhere.
>
> Thank you for the comment. The reviewer is right that $K_Q$ and $K_D$ should be $K$. Note that $K$ denotes an upper bound on the norms of various query and document embeddings. In Lemma C.5, we are working with a general version where we have $K_Q$ and $K_D$ as potentially distinct upper bounds on the norms of query and document embeddings, respectively. However, for the rest of the analysis we **implicitly** assumed $K = max\{K_D, K_Q\}$ for the ease of exposition. We will fix Eq. (25) and use $K$ consistently.
>
> > The last line of Eq. (26) is incorrect; it should be ...
>
> Thank you for the comment. The reviewer is correct in pointing out that last line of Eq. (26) should be $ = \mathbb{E}\big[\big(1 - y - \big(1 - \sigma\big( F(q)^TG(d)\big)\big)\big)f(q)^T g(d)\big]$ and not $ = \mathbb{E}\big[\big(1 - y - \big(1 - \sigma\big( F(q)^TG(d)\big)\big)\big)F(q)^T G(d)\big]$.
>
> We would like to note that this typo does not affect the rest of our analysis as $K$ serves as the uniform bound on the norm of both $f(q)$ and $g(d)$ as well.
>
> > The authors should thoroughly check all formulas
>
> Thank you for the comment. We are confident that our main results are all correct. As per reviewer’s suggestion, we will thoroughly proofread all the technical derivations and fix any remaining typos.
>
> > Section 5, which covers the result analysis, is difficult to follow…
>
> We will go over Section 5 again and apply the writing-related suggestions from the reviewer. In particular, we will a) replace complex sentences with simpler sentences, b) remove unnecessary parentheses, c) simplify table captions, and d) update the BEIR table.
>
> > The author uses too much italics in the introduction, which obscures the key points of the paper.
>
> Thank you for the comment. We will revise the introduction to reduce the usage of italics.
>
> > Please explain Figure 4 in Appendix G.
>
> In this figure, we qualitatively compare the embeddings from the [CLS]-pooled and dual-pooled CE models on a set of 6 questions (listed in the figure). We chose two of each questions to be similar. Each question has a positive document and a negative document. So overall we have 72 question-document pairs: 6 question x 12 documents. With that we compute their embeddings and plot the heat map of pairwise distance matrices. Dark color indicates high values, i.e. points are further, and light color indicates low value, i.e. points are closer.
>
> The pairwise distance matrix from [CLS]-pooled are almost degenerate to two points: one point for all positive query-document pairs and other for the negative pairs. This is also illustrated in the top right of the figure showing embeddings for all positive pairs are closer to each other while negative pairs are clustered differently.
>
> In contrast, for our dual-pooled CE model we see a semantically clustered pattern: similar question-document pairs are clustered together while non-similar question-document pairs are farther.

---

> > ### Comment · Reviewer_QGH5 · 2023-08-17
> >
> > Thank you for your response. The new results of CE to DE distillation seem to be better. However, the gap between the CE teacher and the DE student still remains large - more significant than the gap in DE to DE distillation. As I pointed out before, the reason is that the authors didn't come up with a solution that makes the embeddings from a DE well-distributed. Thus I think the contribution is still limited. I've raised my score from 4 to 5.

---

> > > ### Author Response · Authors · 2023-08-17
> > >
> > > Thank you for the response.
> > >
> > > We have significantly improved the performance of the DE student, surpassing the results of previous papers, and also reducing the CE teacher - DE student gap significantly. For example Table 3 from [1], their student (based on DistilBERT -- the same as ours) shows MRR@10 33.2 distilled from their ensemble teacher (which has MRR@10 39.9), and the gap is 6.7. In contrast, the teacher-student gap for our methods is only  3.1(40.0 vs 36.9).
> > >
> > > Also we believe that there is a big difference between the expressive power of the CE model and DE model and the gap is primarily coming from that. In particular, the late interaction between query and document representation via only a dot product is very limiting. If we keep the same embeddings but replace dot product with a richer late interaction like ColBERT [2], we can eliminate most of the gap (now the gap is only 0.9):
> > >
> > > | Method                                       |  MRR@10  |  nDCG@10 |
> > > |:---------------------------------------------|---------:|---------:|
> > > |						  |          |          |
> > > | Teacher         				  | 40.0     | 45.8     |
> > > | DE student with EmbedDistill                 | 36.9     | 42.6     |
> > > | + **Replace dot with ColBERT**               | 39.1     | 45.0     |
> > > |                                              |          |          |
> > >
> > > This shows the embedding quality might not be the main culprit for the gap.
> > >
> > > [1] Hofstätter et al. "Improving efficient neural ranking models with cross-architecture knowledge distillation." arXiv preprint arXiv:2010.02666 (2020).
> > >
> > > [2] Khattab et al. ColBERT: Efficient and Effective Passage Search via Contextualized Late Interaction over BERT, page 39–48. Association for Computing Machinery, 490 New York, NY, USA, 2020.

---

### Official Review · Reviewer_cm3b · 2023-07-07

**Soundness:** 3 good
**Presentation:** 3 good
**Contribution:** 3 good
**Rating:** 6
**Confidence:** 4

**Summary:**

This paper proposes to learn dense neural IR models by distilling not only the teacher scores but also by aligning the learned representations.

The paper has a theoretic part that motivate the actual embedding distillation with two propositions. The first one shows that the teacher-student gap is bounded by an expression where some terms are the difference between teacher/student embeddings. The second bounds one of term of the expression by leveraging the $u$-covering number of function classes for teacher and student.

Experiments show that using a regularization based on embeddings improve results in two settings: dense to dense and cross-encoder to dense distillation. In the latter case, the authors propose a simple but original approach to get a document and query representation.

The experiments are conducted on the Natural Questions and MS-Marco datasets.



**Strengths:**

This modification of the distillation procedure can easily be applied to all dense model, so the impact of the paper might be important in changing the mainstream procedure.

There is also a nice result showing that just using embedding alignment (without the other part of the distillation loss) words pretty well.

The mathematics justifying the approach are also interesting, by integrating the distance between teacher and student embeddings in the picture, but the bound do not seem to be of practical interest (apart from motivating the approach) and thus might not need to be that central in the paper (esp. proposition 3.2).

**Weaknesses:**

In the experiments, the training based on query generation should not be the last model modification – it should rather the query/document embedding part, since query generation is not the core method proposed by the authors. This would also allow to see what exactly is brought by aligning the embeddings.

The cross-encoder based distillation (which is the main one used by dense models) is quite disappointing in term of performance (even the best CE model performs worse than the best dense model, which should not be the case)… and suggests that to properly train a dense model with their method, one needs a very well trained dense models (but still, distilling to a lower-capacity model is interesting in that case, although the difference between the two procedures is not discussed or experimented with.

The discussion 196-206 is not really needed (it is quite obvious that aligning with the `[CLS]` does not make sense).

**Questions:**

1. Please clarify what is the $u$-covering number and how important it is in the case of the problem

2. what is “inheriting the document tower” (l. 189)

3. How are the CE models trained (this should be in the main text), especially in case of the modified CE models that allow computing meaningful document/query representations.


**Limitations:**

The limitation section is quite generic apart from the first sentence that could be more discussed. Also, the fact the CE distillation is not that performant is (at least for me) a major limitation (since cross-encoders are so much powerful), but it is also a challenge for this type of method.

---

> ### Author Rebuttal · Authors · 2023-08-07
>
> Thank you for your detailed review. Please find our point-by-point response to your comments/questions below.
>
> > The experiments are conducted on the Natural Questions and MS-Marco datasets.
>
> Please note that we also evaluated our proposed approach on BEIR benchmark to measure its utility in zero-shot settings (Table 4 and Appendix F.4).
>
> > In the experiments, the training based on query generation should not be the last model modification....
>
> Thank you for the comment. Please note that the effect of embedding alignment **without query generation** can be viewed from  “+ Query embedding matching” and “Train student using only embedding matching and inherit doc embeddings” rows in Table 1, 3 (corrected version in the appendix), 6, 7, and 9. As evident from these tables, without any query generation the alignment on its own covers most of the gap between the teacher and student models.
>
> > The cross-encoder based distillation (which is the main one used by dense models) is quite disappointing …
>
> For the CE to DE distillation, Table 5 employs a RoBERTa-based model as the CE teacher. This indeed shows poorer performance compared to the DE to DE distillation results in our submission.
>
> That said, we would like to bring Appendix F.3.2 (Table 9) to the reviewer’s attention, where we utilized a much stronger CE teacher obtained from SimLM (Wang et al., ACL 2023), which we were able to obtain between the deadline for the main text and appendix. We will include this new result in the **main body** of the final version of the paper.
>
> With this stronger teacher CE model (MRR@10 of 40.0 nDCG@10 of 45.8), we are able to obtain significantly improved performance for CE to DE distillation via EmbedDistill. In particular, Table 9 shows the utility of four key techniques towards CE to DE distillation: 1) dual pooling; 2) embedding alignment; 3) inheriting document embeddings from the CE model (as enabled by the dual pooling); and 4) query generation. In the revised version, we will include Table 9 in the main text.
>
> Here we copy Table 9 from Appendix F.3.2 for reviewer’s attention.
>
> | Method                                       |  MRR@10  |  nDCG@10 |
> |:---------------------------------------------|---------:|---------:|
> |						    |          |          |
> |Train student directly                        | 27.0     | 32.2     |
> |+ Distill from teacher                        | 33.2     | 38.7     |
> |+ Inherit doc embeddings                      | 35.4     | 41.0     |
> |+ Query embedding matching                    | 36.1     | 41.7     |
> |+ Query generation                            | 36.3     | 42.0     |
> |Only embedding matching and inherit doc embed | **36.9** | **42.6** |
> |+ Query generation                            | 36.8     | 42.5     |
> |                                              |          |          |
> |Standard distillation from [CLS]-teacher      | 32.8     | 38.4     |
>
> Please see Appendix F.3.2 Table 9 for details.
>
> > The discussion 196-206 is not really needed
>
> Thank you for the comment. The discussion might not be obvious to some readers. However, we would be happy to move this discussion to the appendix if the reviewer feels strongly about it.
>
> > Please clarify what is the u-covering number...
>
> $u$-covering is a measure to quantify the complexity of a function class which is an important tool in deriving uniform deviation bounds. It’s utilized in the proof of Theorem 3.2 (see line 786 and Lemma C.5 in the appendix). We will define it either in the main text or in the appendix of the revised version.
>
> > what is “inheriting the document tower” (l. 189)
>
> We believe that the term **tower** is the cause of confusion here. It should have been **encoder**.
> For the broader context, as one of the contributions in our work, we consider **asymmetric** architecture for the student DE model. As an example, for DE to DE distillation, the student simply utilize the frozen (non-trainable) document encoder of the teacher model and we only train the query encoder of the student model along with necessary projection layers to match the embedding dimension of the two encoders (see Figure 1a for an illustration). The procedure of utilizing the frozen (non-trainable) teacher document encoder is referred to as “inheriting the document ~~tower~~ encoder”.
>
> Note that such an asymmetric architecture does not increase test-time latency as document embeddings for a corpus can be constructed offline and only query embedding needs to be generated in an online manner.
>
> > How are the CE models trained (this should be in the main text)...
>
> For the CE model used in Table 5, line 327 – 329 provides a short description of its training objective Appendix  F.1 gives details of the optimization method. Our CE models are trained with the dual pooling scoring loss in order to obtain the query and document embeddings from the same network (line 207-212, and Appendix B). We also found for the SimLM case, a variant of reconstruction loss (Line 214) was helpful to encourage the dual pooling even further. We will consolidate these and present an expanded discussion in the main text of the revised version.
>
> >The limitation section...
>
> Thank you for your comment. We tried to be fair and forward-looking while discussing the limitations of our study. We would be happy to include any specific limitations/topics that the reviewer thinks are relevant in the revised version.
>
> > Also, the fact the CE distillation is not that performant is (at least for me) a major limitation..
>
> As discussed in response to the reviewer’s earlier comment, Table 9 in Appendix F.3.2 presents much stronger results for CE to DE distillation by leveraging SOTA CE model – SimLM (Wang et al., ACL 2023).
>
> **We respectfully hope that this would change the reviewer’s opinion about CE distillation not being performant and the reviewer will reconsider their final rating of our submission.**

---

> > ### Comment · Reviewer_cm3b · 2023-08-14
> >
> > Thanks for your answer and further results related to the cross-encoder to dense-encoder distillation which are indeed interesting. Overall, I think the paper makes a contribution to the information retrieval field – although I agree with the other reviewers this is not a strong contribution.

---

### Official Review · Reviewer_579a · 2023-07-09

**Soundness:** 3 good
**Presentation:** 3 good
**Contribution:** 3 good
**Rating:** 6
**Confidence:** 3

**Summary:**

This work proposed to use embedding matching task for knowledge distillation for IR: unlike in traditional knowledge distillation where the matching score of query and document is used, the embedding matching tasks try to align the embedding representation of query and document. It works both for dual-encoder model (DE) and cross-encoder model (CE). The student model learned achieves 1/10th of size while retraining 95-97% of the teacher performance.

This work presented a theoretical analysis of the teacher-student generalization gap for IR settings. The student DE model proposed has an asymmetric configuration where the query encoder is smaller than document encoder, and the later is frozen during knowledge distillation. This could reduces inference latency at query time.

To validate the effective of the proposed method, experiments on common benchmark for IR is used. The model achieves competitive results on Natural Questions, MSMARCO and BEIR (zero-shot IR benchmark).

**Strengths:**

* In-depth theoretical analysis of the teacher-student generalization gap in IR models, which inspired the embedding alignment methods.
* Extensive experimental results to validate the proposed method: it achieves competitive results on both Natural Questions, MSMARCO datasets and also the BAIR datasets in zero-shot cases.
* Overall well written and easy to follow.


**Weaknesses:**

Overall, the technical contribution presented in this work isn’t super strong in my opinion. Knowledge Distillation is known to be effective for IR, both for DE and CE models. The embedding alignment techniques has also be explored in other settings. Although specific technical challenges needed to be addressed in applying embedding alignment for knowledge distillation of IR models, it does not seem to be groundbreaking.



**Questions:**

* Q1: Has aligning other layers of the network being considered/explored?


**Limitations:**

The authors addressed the limitations of the work where it only studied transformer-based models (but expect it to work for other models such as MLP). It only studied IR in text and did not extend to multi-modal cases.

---

> ### Author Rebuttal · Authors · 2023-08-07
>
> We thank the reviewer for their comments and acknowledging the value of both our theoretical analysis and extensive experimental results. Below, we address your comments in the weaknesses sections and questions.
>
> > …Knowledge Distillation is known to be effective for IR, both for DE and CE models. The embedding alignment techniques has also be explored in other settings. Although specific technical challenges needed to be addressed in applying embedding alignment for knowledge distillation of IR models, it does not seem to be groundbreaking.
>
> We have acknowledged both growing score-based distillation literature in IR settings as well as prior work on **distillation with representation (embedding) alignments** for **non-IR** setups in Section 6 of our submission. As mentioned in our submission, distillation for IR applications presents unique challenges around cross-architecture distillation, dealing with negative sampling, partial alignment, etc. For example, in our work, we explore DE-to-DE distillation while inheriting teacher document encoder and CE-to-DE distillation aided by dual pooling. Such opportunities/challenges are not necessarily present in other domains where distillation is employed. Furthermore, we also accompany our proposed embedding-based distillation for IR with a rigorous theoretical analysis/justification, which to the best of our knowledge is also novel.
>
> > Has aligning other layers of the network being considered/explored?
>
> Indeed, we mainly explored aligning final layer embeddings between the student and teacher encoders in our submission. In our comprehensive experimental studies, such an alignment was already powerful enough to bridge most of the gap between the student and teacher models. Here, we would also like to point out that directly inheriting the frozen document encoder from the teacher can be treated as a stronger form of alignment. Beyond this, exploring alignment of other (trainable) layers of the networks is indeed an interesting direction that we would plan to explore in a future work.
>
> > The authors addressed the limitations of the work where it only studied transformer-based models (but expect it to work for other models such as MLP). It only studied IR in text and did not extend to multi-modal cases.
>
> We focused on transformers as they have increasingly become the *architecture of choice* in various domains of ML, including IR. Since we focus on only aligning final layer embeddings, we expect our techniques to work for other architectures as well such MLP-based encoders. Regarding exploration of multi-modal settings, we do agree that it’s an interesting direction for future work.

---

> > ### Comment · Reviewer_579a · 2023-08-21
> >
> > Thank the authors for the details responses. I've read the authors comments as well as other reviewer's feedback and discussions. I decide to keep the original score.

---

### Official Review · Reviewer_KiMa · 2023-07-10

**Soundness:** 3 good
**Presentation:** 3 good
**Contribution:** 3 good
**Rating:** 5
**Confidence:** 5

**Summary:**

The present study introduces a new technique, EmbedDistill, designed to transfer knowledge from large-scale neural networks to smaller versions for information retrieval (IR) purposes. This approach utilizes the relative geometric connections between queries and documents, as learned by the more extensive teacher model, to synchronize the representations of both teacher and student models. Furthermore, it scrutinizes the data manifold to diminish disparities between the student and teacher, particularly in areas where the training data is scant. The findings suggest that this proposed method can effectively distill both dual-encoder and cross-encoder teacher models into 1/10th size asymmetric students, maintaining 95-97% of the original teacher's performance. The study's key contributions consist of a groundbreaking geometric distillation technique, a unique query creation method for enhancing distillation quality, and practical results underlining the success of the proposed method.

**Strengths:**

The paper presents a new approach to distilling knowledge from strong neural models to weaker ones for information retrieval. The paper makes several contributions that are noteworthy.

- The paper's approach is original in that it leverages the relative geometry among queries and documents learned by the large teacher model to align the representations of the teacher and student models.
- The paper is of good quality, with a well-defined problem statement, clear methodology, and adequate experimental evaluation. The paper's contributions are remarkable in that they provide a new approach to distilling knowledge from strong neural models to weaker ones for information retrieval. The authors provide a detailed analysis of the proposed approach and compare it with existing methods. The experiments are well-designed and the results are presented in a clear and concise manner.
- The paper is well-written, well-organized, and easy to follow. The authors provide clear explanations of the proposed approach and the experimental setup.


**Weaknesses:**

Although there are several distillation methods proposed in this paper, the core idea of this paper is to distill representation, via L2 distance, from a strong representation model to smaller ones. And the derivation of the representation is also straightforward and brute-forcing, i.e., CLS representation or mean pooling over corresponding text.

The backbone of the retriever is too weak, which cannot verify the generality of the proposed methodology. That is, it’s required a stronger student model to check if the validity of proposed methodology will be mitigated with stronger students. Meantime, the neural retrievers and rankers used in this work is relatively out-of-date. It’s recommended to leverage more state-of-the-art retrievers/rankers to make the experiments more convincing.


**Questions:**

Refer to the Weaknesses

**Limitations:**

Yes

---

> ### Author Rebuttal · Authors · 2023-08-07
>
> Thank you for your positive feedback. Below we aim to answer your questions raised in the *weaknesses* section.
>
> > Although there are several distillation methods proposed in this paper, the core idea of this paper is to distill representation, via L2 distance, from a strong representation model to smaller ones. And the derivation of the representation is also straightforward and brute-forcing, i.e., CLS representation or mean pooling over corresponding text.
>
> We agree that aligning the teacher and student embeddings via L2 distance after suitable projection is the core idea. Note that this simple idea is grounded in theory and highly effective as demonstrated by our comprehensive experimental results in the main body and appendix.
>
> We would also like to bring the reviewer’s attention to novel asymmetric configuration (cf. lines 179 -- 191) for student DE models where we simply inherit the frozen document encoder from the teacher without increasing the test-time latency of the student model. Furthermore, training a CE model with dual pooling in order to realize improved distillation from CE to DE model (cf. line 207 -- 219 in the main text and Appendix F.3.2) is another novel contribution of our work.
>
> > The backbone of the retriever is too weak, which cannot verify the generality of the proposed methodology…Meantime, the neural retrievers and rankers used in this work is relatively out-of-date
>
> Please note that we utilize Adversarial Retriever-Ranker (AR2) model (Zhang et al., ICLR 2022 [1]) as the teacher for DE to DE distillation on NQ dataset. Its recall@100 performance is close to the SoTA method (see Table 2 in the main text and open leaderboards e.g. PaperWithCode). Similarly, our DE teacher model on MSMARCO dataset – SenctenceBERTv5 (updated in 2021) has a very competitive performance in terms of MRR@10 and NDCG@10.
>
> As for the CE teacher on MSMARCO dataset, we utilize the dual-pooled version of a very recent and strong CE model – SimLM (Wang et al., ACL 2023 [2]) – in Appendix F.3.2.
>
> Our proposed EmbedDistill approach consistently leads to improved performance with a wide range of aforementioned teacher models. In addition, we had also explored some in-house (weaker) teacher models on NQ and MSMARCO in Appendix F.5. Again, EmbedDistill **consistently** showed improved performance.
>
> The main goal of distillation is to bridge the gap between the teacher and student model with as small a student as possible. We are able to attain this goal with a very simple DistillRoBERTA/DistillBERT and DistillRoBERTA-mini/DistillBERT-min architectures, which again alludes to the strength of our proposed approach.
>
> We believe that the comprehensive positive results on a wide range of teacher architectures does clearly establish the utility of EmbedDistill. That said, if the reviewer has some specific model architecture in mind, we would be happy to try to include the results with those in a revised version.
>
> [1] Hang Zhang, Yeyun Gong, Yelong Shen, Jiancheng Lv, Nan Duan, Weizhu Chen, “Adversarial Retriever-Ranker for Dense Text Retrieval”, ICLR 2022
>
> [2] Wang, Liang, Nan Yang, Xiaolong Huang, Binxing Jiao, Linjun Yang, Daxin Jiang, Rangan Majumder, and Furu Wei. "Simlm: Pre-training with representation bottleneck for dense passage retrieval." ACL 2023

---

### Decision · Program_Chairs · 2023-09-21

**Decision:**

Reject

**Comment:**

This paper proposes a knowledge-distillation method for IR tasks. The authors use embedding matching & query generation to align & reduce teacher-student discrepancy. Overall, the paper is motivated well & addresses an important problem. While the theoretical analysis of the teacher-student generalization gap is interesting and the paper is written well, the contributions & presented results, as of now, still requires some work to address some of the reviewer concerns and gaps. The authors did provide clarifications in their rebuttal and subsequent engaging discussions on many issues which is helpful & appreciated and also addresses some of the reviewer concerns. However, there are a few aspects that still require more attention & analysis. For example, results reported in the CE to DE setting (Table 5) are not comprehensive and while the authors reference the appendix (and other results), this is not enough to validate the efficacy of the approach. What can help significantly strengthen the paper is more detailed experiments and comparisons against baselines (not just ablation studies for the proposed method) for the different distillation settings and ideally, on other IR datasets/tasks as well. The teacher model choice (& capacity) seems to have a huge effect on the final performance, so adding ablation along teacher dimension (architecture, etc.) can shed some light on where this particular approach can be effective vs not.

Overall, the paper presents interesting work and a useful direction to explore; the authors are encouraged to use the feedback to further strengthen the paper for a future submission.